# The Astrobiology of Alien Worlds: Known and Unknown Forms of Life

**Louis N. Irwin** [1] and **Dirk Schulze-Makuch** [2,3,4,5,*]

[1]  Department of Biological Sciences, University of Texas at El Paso, 500 West University Avenue, El Paso, TX 79968, USA; lirwin@utep.edu

[2]  Astrobiology Research Group, Center for Astronomy and Astrophysics (ZAA), Technische Universität Berlin, Hardenbergstr. 36, 10623 Berlin, Germany

[3]  Section Geomicrobiology, German Research Centre for Geosciences (GFZ), 14473 Potsdam, Germany

[4]  Department of Experimental Limnology, Leibniz-Institute of Freshwater Ecology and Inland Fisheries (IGB), 12587 Stechlin, Germany

[5]  School of the Environment, Washington State University, Pullman, WA 99163, USA

*  Correspondence: schulze-makuch@tu-berlin.de; Tel.: +49-30-314-23736

**Abstract:** Most definitions of life assume that, at a minimum, life is a physical form of matter distinct from its environment at a lower state of entropy than its surroundings, using energy from the environment for internal maintenance and activity, and capable of autonomous reproduction. These assumptions cover all of life as we know it, though more exotic entities can be envisioned, including organic forms with novel biochemistries, dynamic inorganic matter, and self-replicating machines. The probability that any particular form of life will be found on another planetary body depends on the nature and history of that alien world. So the biospheres would likely be very different on a rocky planet with an ice-covered global ocean, a barren planet devoid of surface liquid, a frigid world with abundant liquid hydrocarbons, on a rogue planet independent of a host star, on a tidally locked planet, on super-Earths, or in long-lived clouds in dense atmospheres. While life at least in microbial form is probably pervasive if rare throughout the Universe, and technologically advanced life is likely much rarer, the chance that an alternative form of life, though not intelligent life, could exist and be detected within our Solar System is a distinct possibility.

**Keywords:** definition of life; exoplanets; alien life; habitability; planet; moon

## 1. Introduction

The abundance of life on Earth shows that our planet's geophysical characteristics are quite conducive for life as we know it. However, our knowledge of planets and moons other than Earth in our Solar System—and what we are rapidly learning about planetary bodies in other solar systems—tells us that the geophysical characteristics of our planet are relatively rare. It could be that the form of life we know is the only one possible, due to the universal physicochemical constraints on the nature of complex systems, and that life flourishes on Earth because our planet is particularly amenable to those constraints. If so, then life on other worlds is likely to be very rare and not unlike life on Earth. However, there is no logical reason to assume that the laws of physics and chemistry and the nature of complex systems limit the possibilities for the nature of life only to the characteristics of living systems with which we are familiar. Therefore, life on other worlds could be radically different from life as we know it. Even if living systems universally are bound by structural and metabolic constraints at the most reductionistic level (comparable to cells as we know them), organisms at the macroscopic level in very different environments may take on forms that are either unknown, obscured, or very rare on Earth—as broadly acknowledged in the current field of astrobiology [1].

This review considers the different forms of life that might exist on other planetary bodies, given the past and current geophysical conditions on those worlds. We first discuss the problem with, and the necessity for, a clear definition of life. We then review the basic characteristics of life as we know it, highlighting some unusual or lesser known forms it has taken on Earth, followed by a brief review of radically different forms that have been imagined or demonstrated experimentally to a degree. Finally, we consider the possible forms of life that would be most plausible on the different types of planetary bodies of which we are aware.

## 2. The Definition Issue

The search for life on other worlds thus requires us to have a generic definition of what we are searching for—a definition that captures the essence of the phenomenon of life without tying it to characteristics that may be peculiar to life on Earth. The attempt to come up with such a definition does not lack for historical effort [2–8]. One ambitious volume [9] begins with different definitions from 79 authors.

### 2.1. Common Components of Modern Definitions

While definitions differ greatly, certain elements often recur. They include (1) encapsulation, (2) complexity, (3) thermodynamic disequilibrium, (4) energy consumption, (5) chemical dynamism, (6) self-organization, (7) homochirality, (8) selective exchange of materials, (9) environmental sensitivity, (10) homeostasis, (11) autocatalysis, (12) reproduction, (13) adaptive adjustment, (14) coding for retention of form and function across generations, (15) teleology, and (16) evolution. Many variants and elaborations of each of these terms are commonly advanced. Several of them are controversial—especially the last two. Every one of them includes examples that all would agree can be found in the non-living world.

### 2.2. Argument against a Definition

In the face of many counterexamples, and absent, in their view, a satisfactory theory for the nature of living systems, Cleland and Chyba [10] argued that disagreements over the definition of life were likely to be inescapable and interminable. Cleland [11] continued to assert that scientific theories are not the sort of thing that can be encapsulated in definitions, and since "scientists are currently in no position to formulate even a tentative version of such a theory", neither a definition nor theory of life is possible at this time. More recently, Cleland [8] has sharpened her critique of efforts to define life on grounds that (1) most such efforts relate to an antiquated Aristotelian biology, and (2) reliance on life only as it is manifested and understood on Earth is likely to be unrepresentative of other possible forms of life. She argues that the search for life on other worlds should therefore abandon reliance on seeking detection of defining characteristics and focus instead on finding "potentially biological" anomalies [12] worthy of further examination.

### 2.3. Necessity for a Definition

We agree with Cleland and her like-minded colleagues that life on Earth may be unrepresentative of life on other worlds—maybe on most other worlds—and, therefore, that reliance on the characteristics of terran life for defining life in general is inappropriate. We do not agree, however, that a generic theory of life is unachievable. We do not even agree that a consensus on the definition of life, broadly speaking, is lacking. Taken as a whole, the modern literature on generic characteristics of living entities shows broadly held acceptance of at least three features: (1) boundary conditions that encompass an internal environment at a lower state of entropy and therefore in thermodynamic disequilibrium with the external environment; (2) influx of energy from the external environment to maintain the low entropic state internally and do work; and (3) the ability to self-reproduce.

Many authors, including NASA [13], add a fourth feature: the ability to undergo Darwinian evolution. While any form of life obviously must evolve from non-living precursors, and we agree

with NASA that this is a feature of life as we know it, we do not consider it useful as a generic criterion in the search for alien life. First, it can only be gauged over multiple generations and, therefore, is not useful for assessing whether individual entities are alive. Secondly, "Darwinian evolution" implies change by natural selection which is not the only mechanism for biological innovation, and even in some terran forms occurs barely at all over extended periods.

Faced with the prospect of alternative forms of life in a variety of different geophysical settings, any putative form of life encountered could be totally unfamiliar. Therefore, basic characteristics of living entities need to be specified. A theory of life based on the three fundamental features above provide objective criteria for detecting life anywhere [6,7,14]. All three characteristics are necessary, but if all three are present, they are sufficient to qualify an entity as being alive.

As a practical matter, it may be difficult to detect one or more of these characteristics, especially remotely. Therefore, the strategy of using "tentative criteria" to search for "possibly biological" anomalies in candidate entities makes sense [12], bearing in mind that any characteristic deemed "possibly biological" requires a definition for "biological" and that a judgment about which criteria would be appropriate even tentatively requires some notion about a theory of life.

In the sections to follow, by "life" we mean physical entities that possess the three fundamental features defined in the first paragraph of this subsection.

## 3. Life as We Know It

### 3.1. Biological Form and Function as Commonly Observed on Earth

Life on Earth is the only life we know. It is closely associated with the natural history of our planet, which formed approximately 4.6 billion years ago. Earth, like the other terrestrial planets—Mercury, Venus, and Mars—formed from protoplanets that accreted relatively close to the Sun, from mostly rock after much of the gas evaporated. Impacts were common during Earth's early planetary history and seem to have subsided in frequency approximately 3.8 billion years ago [15]. Life appears to have been present soon after the end of this period [16,17]. Mineralogical evidence points to the existence of life by 4.1 billion years ago [18] with fossil evidence of life at least 3.8 billion years ago [19]. Thus, it appears that life arose rather quickly, somewhere between 4.4 and 3.8 billion years ago.

We do not know where life originated, but its ionic composition mirrors that of sea water to a large extent, which supports the origin of life in marine water on Earth rather than seeding by transport from another location [7]. Strong cases have been made for a suboceanic origin of life, particularly at alkaline hydrothermal vents [20–23]. An alternative argument has been made for an origin in continental hydrothermal springs [24,25]. Both of these scenarios would be consistent with an origin in water. However, other locations for the origin of life have been advanced as well [26–29].

Water is the critical solvent for life on Earth [7]. Most biomass on our planet is phototrophic. Photosynthesis can be either anaerobic, which does not produce oxygen as a byproduct, or oxygenic which does produce oxygen and is responsible for the oxygen-rich atmosphere on Earth. Anaerobic photosynthesis is still used by many microbes on Earth, while cyanobacteria and some other microbes and all plants use oxygenic photosynthesis. The other main metabolic strategy on Earth is chemotrophy, which can be either autotrophic (deriving energy from redox reactions with inorganic substrates) or heterotrophic (using organic substances as a food source).

Life retains its species–specific characteristics through cycles of reproduction and regeneration by encoding information in nucleic acid polymers, from which proteins are translated with different functions based on conformational specificity. The coding polymer for genetic information is almost universally DNA, which is based on nucleotide base triplets (codons). A total of 64 of these codons are assigned to 20 amino acids with the remaining three serving as stop codons. This arrangement is known as the standard genetic code because of its (1) strict codon symmetry; (2) specification of a definite number of amino acids that serve as protein building blocks; and (3) redundancy of physico-chemical properties, especially with regard to the hydrophilicity and hydrophobicity of different amino acids [30].

The universality of the code for life on Earth allows horizontal gene transfer—probably one of the most powerful tools, particular in microorganisms, for adapting to new environmental challenges. Although universal, the code has some variability and flexibility. For example, some codons have their meanings re-assigned in specific organisms, particularly those that reside in biological niche environments. Also, some organisms naturally encode for two additional amino acids: selenocysteine and pyrrolysine [31]. Despite these variations, all organisms have three nucleotides in their codons and utilize tRNA and ribosomes to read the code in the same direction for translating it into a specific sequence of amino acids. While DNA is used by all organisms on our planet, some viruses use RNA instead, which likely served as the coding polymer prior to DNA during the hypothesized RNA world.

At the phenotypic level, many different schemes are used for reproduction. Binary fission is the general mechanism used at the cellular level, but budding, regeneration, and vegetative extension are also common. Most, but not all macroscopic, complex life use sexual reproduction at some phase in the life cycle.

The vast majority of organisms on Earth are unicellular and microscopic as they have been throughout their evolutionary life span. However, a distinct fraction of life is multicellular and macroscopic. The overall complexity of life on our planet has inevitably increased, but as the exception rather than the rule within most taxonomic groups [7,32]. There is also an amazing degree of symbiosis among different species, sometime so intrinsic that it is difficult to decide where one species begins and the other ends (as in lichens). Life has occupied nearly every potential habitat on our planet where water can remain in liquid form below the upper thermal limit for viability which is likely approximately 150 °C [33]. Biomass and biodiversity are greatest in the subterranean and aquatic environments (both fresh and saltwater), while plant biomass dominates the surface of the land [34] which was occupied later in the natural history of our planet as co-evolution of protection against UV irradiation and dehydration made it habitable. The atmosphere serves as a temporary habitat by all the different species that rely on flying as a major means of locomotion and by an overwhelming number of microbes and other light organisms that use the wind and atmospheric currents for dispersion.

The diversity of life on our planet is staggering. It arises from the environmental heterogeneity of our planet, with all its niche fragmentation due to the different kinds of substrates and environmental and climatic conditions. This, in turn, has resulted in intricate food webs and multiple trophic levels. While individual organisms are quite fragile, life itself is quite hardy, as evidenced by the fact that life as we know it has survived for approximately 4 billion years despite numerous global environmental challenges and natural catastrophes.

### 3.2. Amorphous Organic Forms

Our conventional view of living organisms is that of a unicellular microbe or a multicellular entity more-or-less constant in shape and size (once fully developed). Not all forms of life as we know it, however, conform strictly to this model.

A well-known example of amorphous organic organisms on Earth are the slime molds which, under certain conditions, aggregate into acellular, multinucleate bags of protoplasm of indefinite shape which move through cytoplasmic streaming [35]. They reproduce sexually via several different mating types [36], but in the plasmodial (multinucleate unicellular) state they may break apart into unequal fragments which can also remerge into a single organism. A more exotic example has been the discovery of massive fungi of the genus *Armillaria* [37,38]. The largest yet detected is a single organism extending over 8.8 km$^2$ of forest floor with an estimated age of about 2500 years [39]. These terran organisms reproduce sexually but may expand asexually over large areas and are remarkably resistant to genomic change [40]. Other planetary bodies, where conditions may be more uniform and stable, like subsurface oceans, could favor the existence of extended, slowly metabolizing, asexual organisms such as these.

Another, more complex example of amorphous organisms are clonal plants. While reproducing sexually, they often expand primarily by asexual vegetative generation of new bodies (ramets). Thus,

linked to the structures that gave rise to the new outgrowths, they constitute a single organism. One of the most familiar examples is *Populus tremuloides*, the North American aspen [41]. Genetically identical members of this species are long-lived and can expand over a large area with hundreds of clonal outgrowths. Under uniform and invariant environmental conditions, long-lived, asexual expansion of single amorphous individuals could be favored.

### 3.3. Amorphous Conglomerate Forms

A symbiotic aggregate of two or more different types of organisms, or a conglomerate of non-living and living components, can give rise to amorphous structures not readily identifiable as living organisms. Awareness of these biologically generated forms is important for detecting unfamiliar life on other worlds.

The simplest, familiar aggregate of this type on Earth is a lichen—a combination of algae or cyanobacteria and fungus [42]. The symbiotic relationship between the autotrophic algae or bacteria and the heterotrophic fungi gives the combination the ability to grow on solid surfaces in a great range of environments.

Microbialites are microbially induced sedimentary structures [43]. The oldest microbialites are thought to be microbial mats, which are dynamic and complex ecosystems consisting of heterogeneous microbial populations, often segregated spatially and temporally into distinct microenvironments [44]. Microbial mats are believed to have been widespread on the bottom of lakes and shallow seas prior to the evolution of burrowing organisms that broke up their tight-knit structure, promoting their demise from oxygen toxicity. Stromatolites appeared at ~3450 Ma and were generally diverse and abundant from 2800 to 1000 Ma [45]. The wide range of metabolic processes carried out within different layers contribute to biogeochemical cycles that produce important end products, such as trace gases and mineral precipitates, imparting a "biosignature" preserved in the rock record [45]. These fossil biosignatures of ancient life on Earth could serve as an indication of life on other planetary bodies as already claimed for Mars [43,46,47].

Modern microbialites are found today in the form of large domes and columns with laminated (stromatolite), clotted (thrombolite), and other macrofabrics which may be either agglutinated or mainly composed of calcified or spar-encrusted microbes [45]. We have studied a cold, freshwater aggregate of microbialites in British Columbia in detail, suggesting that they may represent an alternative trajectory toward a community aggregate that achieves some of the biological functions realized by multicellular macroorganisms [48]. This would include, for instance, cellular specializations for certain physiological functions, and the ability to grow larger for defensive purposes. On planetary bodies, where conditions do not favor the evolution of complex macroorganisms [6,49], this amorphic aggregate of microorganisms may achieve some of the benefits that favor the evolution of larger forms of life under more favorable conditions.

By far the largest amorphous forms of life on Earth are coral reefs. They compose a conglomerate ecosystem based on a mineralized substrate of sessile marine invertebrates of the Class Anthozoa in the Phylum Cnidaria [50]. Most corals host symbiotic photosynthetic dinoflagellates, which give corals a diversity of colors and generate the organic nutrients on which the host organism depends [51]. Corals secrete calcium carbonate exoskeletons which, following their breakdown, can become cemented residue adding to the structural substrate on which a great variety of other animals and plants become attached. The resulting local ecosystem is among the most complex found anywhere on Earth.

The substrate of a coral reef is made up of the mineral deposits of clusters of genetically identical polyps which reproduce asexually by budding [52]. Thus, the body of the substrate expands as a contiguous mass. At a distance, coral reefs appear as large, extended, continuous organisms, which in some cases cover vast areas of shallow sea. Indonesia has over 51,000 km$^2$ of coral reefs, while more than 48,000 km$^2$ of reefs line the coastline of Australia [50]. In fact, however, coral reefs are built up from small multicellular animals at a complex stage of evolution. While structures like coral reefs would be easier to detect than other forms of life on other worlds because of their size (provided the

observer has very high spatial resolution), if they do consist of ecosystems composed of a variety of complex macroorganisms, such as coral reefs do on Earth, they are unlikely to have evolved on other planetary bodies unless there had existed conditions conducive for an evolutionary trajectory of complex macroorganisms.

## 4. Known Trajectory of Life on Earth

While we do not know when, where, or how life originated on Earth, we propose from the definition we have adopted (Section 2.3) that it exhibits at least three features: (1) boundary conditions that encompass an internal environment at a lower state of entropy and therefore in thermodynamic disequilibrium with the external environment; (2) use of energy from the external environment to maintain low internal entropy and do work; and (3) the ability to self-reproduce. The challenge at the dawn of life was harvesting the energy for building large organic macromolecules and keeping them from falling apart, encoding the information for their reliable replication [53], and consistently enclosing them apart from their environment [54]. The sequence in which all three elements of a living organism—an enclosed architecture, a proficient energy metabolism, and a genetic code with translational capability—were combined is unknown. Nevertheless, this is our starting point, typically referred to as the Last Universal Common Ancestor (LUCA). This LUCA presumably did not use DNA for replication in the beginning but some other nucleic acid. It also may not have used lipids as semipermeable membranes for encapsulation but some other chemical compounds or minerals. However, it must have had the features discussed above.

Microbial life on Earth has remained essentially in its ancestral form, evolving little since its origin. However, most multicellular forms have evolved through key innovations toward larger and generally more complex organisms. Bains and Schulze-Makuch [55,56] suggested three major paths to innovation: a Critical Path Model, where each major event or innovation requires preconditions that take time to develop (e.g., air bladders before lungs); a Random Walk Model, where each innovation is highly unlikely to occur at any specific time or step, and the likelihood does not change (substantially) with time (e.g., membrane encapsulation of controlled metabolic pathways); and a Many Paths Model, where the major event or innovation requires many random events to create a complex new function, but many combinations of these can generate the same functional outcome, even though the genetic or anatomical details of the different outcomes are not the same (e.g., ability of bats, most birds, many insects, and certain fish to fly). A fourth process, termed "Pulling Up the Ladder", refers to an innovation through either a Critical Path or a Many Paths process that results in destruction of the preconditions for its own innovation.

By analyzing the different key innovations in the evolution of life on Earth, Schulze-Makuch and Bains [56] found that most of them were likely Many Paths processes. If correct, this model predicts that for any evolutionary trajectory, as soon as appropriate preconditions are met, key innovations occur relatively soon, and are likely to occur eventually on all occasions in which the preconditions are satisfied. However, most importantly, if an innovation occurs more than once through a Many Paths process, it is likely to use different mechanisms each time it occurs. Thus, different biochemistries and alternative trajectories for life could arise, even if the "tape of life" on Earth or any other world were replayed.

The first type of metabolism on Earth was presumably chemoautotrophy [57], and many organisms on Earth still use this pathway, most notably at suboceanic hydrothermal vents. An alternative hypothesis, not so much in favor anymore, is that the first organisms might have been heterotrophic chemotrophs, feeding on organic compounds contributed from meteorites and abiotic organic synthesis reactions [58]. Even if so, life must have broadened its metabolic repertoire quickly to avoid starvation. Photosynthesis for converting light energy into chemical energy is usually assumed to have evolved later due to the fact of its more complex biochemistry [59]. However, some sulfur bacteria apparently can carry out both chemosynthesis and (anaerobic) photosynthesis [60] which may hint at a pathway for how photosynthesis originated. It is generally assumed that this first

key innovation of life, photosynthesis, evolved to protect organisms from light and, particularly, from harmful UV radiation [61,62]. Schulze-Makuch and Bains [56] argued that the emergence of photosynthesis was a Many Paths process, because life on Earth has evolved different ways to capture light. Chlorophyll-based photosynthesis is most common, but light can also be captured by rhodopsins or carotenoids and turned into chemical energy. Especially bacteriorhodopsin-based light capture is chemically completely different from chlorophyll-based capture.

Another critical key innovation was oxygenesis, the photosynthetic pathway that produces oxygen as a by-product. It evolved early in life's history on Earth and is likely to have evolved through a series of steps rather than requiring a single huge innovative step. It seems to have appeared first in cyanobacteria. Though highly successful as an energy-gathering pathway, it produced the first global catastrophe for other forms of life on Earth, which were still strictly anaerobic and subject to oxygen toxicity. This Great Oxidation Event occurred about 2.4 billion years ago when Earth's atmosphere (and sea water) reached significant levels of oxygen. Only those organisms capable of mitigating the toxicity of oxygen by evolving oxidation-reduction metabolism survived, thereby altering the overall biosphere. Oxygenesis is clearly a "Pulling Up the Ladder" event, because the results of the innovation destroyed the preconditions for its own occurrence.

The next critical key innovation for life on Earth was the emergence of the eukaryotic cell, hypothesized to have formed from the fusion of a eubacterium with the nucleus of an archaeon [63]. The underlying process of endosymbiosis has been a frequent event that is still occurring today. Chloroplasts in plants and mitochondria in other eukaryotes are also believed to have formed through endosymbiosis. Endosymbiosis occurs among all different types of species, even archaea and animals, and may have evolved from parasitism or from incomplete digestion or both [64]. It is clearly a Many Paths process.

While binary fission is the dominant mode of reproduction for microbes, most eukaryotic cells adopted sexual reproduction as a powerful way to promote genetic variability. In sexual reproduction two cells with one set of chromosomes each (haploid cells) fuse together, resulting in *two* copies of all the chromosomes (diploid cell). Homologous chromosomes (those carrying genes for the same traits) then pair up and swap segments through crossover recombination, generating new permutations of genes which help overcome mutational damage, avoid parasites, and generate new traits for natural selection to act upon. Despite the advantages of sexual reproduction, generally, some species can do without sex, either altogether, such as the bdelloid rotifers, or through parts of their life cycle, like many plants and even some vertebrates. At the other extreme, some fungi show an enormous range of sexual reproductive arrangements with species showing between 1 and 12 mating types ("genders"). Thus, while binary sexual reproduction is a common means of ensuring genetic exchange, it is just one of several possible such mechanisms.

Many distantly related branches of the eukaryotic tree of life hold multicellular life forms, which probably originated from colonial organisms that cooperated closely and adopted multicellularity as a strategy to survive prolonged low-fitness periods [65]. While the achievement of obligate multicellularity with clearly differentiated cell types occurred over millions of years, laboratory experiments show that the first crucial steps in the transition from unicellularity to multicellularity can occur within a couple of days [66]. Furthermore, some branches of the eukaryotes include both unicellular and multicellular species, not only at the phylum level, but also much lower in the taxonomic hierarchy. For example, the yeast family Saccharomycetaceae includes *Saccharomyces cerevisiae*, the yeast famous for making bread, beer, and wine, is clearly unicellular, while the quite closely related filamentous cotton pathogen *Eremothecium gossypii* is an obligate multicellular organism. *Candida albicans* belongs to the same family and is capable of switching rapidly and reversibly between unicellular and multicellular lifestyles. Thus, while complex morphological innovations require more time, and the spread of obligatory multicellularity on a global scale required a very long time historically on Earth, it may have been preceded by experimentation along Many Paths toward multicellularity that continues to the present day within some taxa [56].

A relatively new innovation in the history of life on Earth has been the evolution of larger and more complex fungi, plants, and animals. Two preconditions had to be fulfilled for the emergence of these macroscopic forms: (1) a sophisticated genetic control system and (2) oxygen to power their high-energy metabolism, particularly needed for animal motility (the only possible alternative, fluorine, being too scarce and too reactive) [64]. Due to the predator–prey relationships, multicellular life was accompanied and enabled by increasing size. Another effect of predator–prey relationships was the race for cognitive enhancement between predators and prey. This resulted in the evolution of a few groups of abnormally intelligent species on Earth, including cephalopods, certain birds, such as crows and parrots, cetaceans, and primates.

One of these intelligent animals, *Homo sapiens*, has evolved the cognitive and technological ability to modify its environment to its needs, colonize nearly any habitable stretch of land on our planet, domesticate other animals, rise to a population of billions of individuals, and develop the technology for venturing into space. The remarkable ascendancy to ecological dominance by humans, however, is the exception rather that the rule. As pointed out before, the majority of the biomass on planet Earth for most of its history has been microbial, and even today most of the biomass and diversity resides in either microbes or plants.

## 5. Alternative Forms of Life

Organisms with different chemical compositions and alternative metabolic systems have been proposed to meet the requirements of environments that differ from the liquid water, carbon-rich, chemically reducing, and light-bathed environments in which terran life evolved. The LUCA for forms of life on worlds with conditions drastically different from those on Earth may have been composed of very different chemistries and components. Furthermore, several forms of life not usually considered individual living entities as we know them have been envisioned that stretch but do not contradict the generic features of life described in Section 2 above. All of these must be considered as candidates for alternative forms of life on other worlds, not only because of the radically different conditions from Earth that are likely to be found there, but because an independent origin of life resulting from a different "random walk" [56] may have set in motion a very different trajectory for descendants of that origin. Four different alternative candidates for living entities will here be considered: alternative biochemical systems, unbounded inorganic forms, amorphous organic forms, and mechanical entities.

### 5.1. Alternative Biochemistries

Light and redox reactions are the basic energy sources for life on Earth. However, other energy sources could potentially be tapped, especially in habitats where light is not available such as the ice-covered subsurface ocean environments described in Section 6.1. In those environments, thermal energy, kinetic energy, and osmotic/ionic gradients could be alternative choices [67]. Magnetic energy, for example through the Lorentz force, would also be a possibility, but too little energy could be extracted compared to other available sources. However, magnetic energy may be a feasible energy source on a planet in orbit around a neutron star, given the extremely strong magnetic fields these stars tend to have [68]. Other energy sources, such as gravitational energy, tectonic stress, pressure gradients, and even spin configurations, have been suggested as well, but these energy sources are highly unlikely to be competitive with other forms of energy [7].

There are two energy sources that need special emphasis. The first is thermal energy which could be extracted from the high heat capacity of water or via thermal gradients. Muller [69,70] suggested that thermosynthesis might have been a progenitor of bacterial photosynthesis in the evolution of life on Earth. If so, thermal energy might be the energy source that we would expect most to be tapped in environments where no light is available. In principle, more energy could be extracted from thermal energy than either light or chemical energy, but the underlying problem is the 2nd Law of Thermodynamics, which limits the amount of usable energy (a large fraction of that energy is lost to entropy). Light is a great energy source for surface life on Earth and is used to produce by far the

largest amount of biomass, although only a few narrow bands of the visible light frequencies are used. Life as we do not know it may use other frequencies that depend on the type of host star, the planetary body's atmosphere, and the biochemistry of the alien life. However, much lower frequencies and less energetic electromagnetic waves would likely not be sufficient to power life, but may only be used as a supplemental energy source, as suggested by [71]. Higher frequency electromagnetic waves, like UV radiation, X-rays, or even ionic radiation, would be extremely challenging for the biochemistry of any life to harvest. The high amounts of energy inherent in these types of radiation would likely destroy the organism. In the case of ionic radiation, an additional problem is that radioactive decay is a stochastic process that does not allow for the predictable electromagnetic quanta so suitably used by life on Earth during photosynthesis. Nevertheless, it has been suggested that certain organisms (specific fungi) might be able to harvest ionizing radiation despite those challenges [72].

While all life on Earth uses redox reactions for metabolism, different types of chemical reactions could be used as energy sources in starkly different types of environments. For example, Schulze-Makuch and Grinspoon [73] suggested the use of radicals for putative life on Saturn's moon Titan. While radical chemistry would be generally too energetic for life's biochemistry on our planet, it might be appropriate for overcoming kinetic barriers existing in an extremely cold environment. On Titan, for example, there are liquid hydrocarbons on the surface of Titan at less than 100 K (Section 6.3), which could take the function of liquid water as a mediator of chemical reactions. Most of the potentially alternative solvents for life are liquid at lower temperatures than water. Of these choices, ammonia or ammonia–water mixtures are probably the most promising, particularly because ammonia could easily interact with amino acids which are the basic building blocks of life on Earth. Polar hydrocarbons would have potential as well, because of their ability to enhance organic synthesis reactions. Methanol, for example, has a larger liquidity range than water (−94 °C to 65 °C under Earth surface conditions), promotes the formation of sugars [74] and mixes well with water.

Carbon is the major building block of life on Earth and its versatility to form millions of complex polymers encompasses most of known organic chemistry. Due to the favorable properties and versatility of carbon, we would also expect unknown types of life to be based on carbon as a major element. Only one other element, silicon, might be able to replace carbon in its prominence under some very specific environmental conditions. Silicon can also form organic compounds which have also been detected in space. Under Earth conditions, however, silicon reacts both with water and molecular oxygen, and is quickly immobilized as silicates. Petkowski et al. [75] explored silicon's chemical complexity in several solvents present in planetary environments such as water, cryosolvents, and sulfuric acid. They concluded that in no environment would life based primarily on silicon chemistry be a plausible option. Surprisingly, however, sulfuric acid appeared to be able to support a much larger diversity of organosilicon chemistry than water. Silicon would also fit chemically with a hydrocarbon solvent [7]. It is noteworthy that silicon plays a minor but significant role in life on Earth (e.g., shells of diatoms consist entirely of silicon dioxide) and could potentially play an even more important role in an alien biosphere. In principle, what the specific elements or molecules would be in an alien biological scheme is less pertinent than the specific functional properties they would provide. We have to realize that our knowledge is extremely limited, based on only one biosphere and one biochemistry of life (Section 3.1). Although this known biosphere is very diverse, we probably extremely underestimate the forms and functions life can take.

Intriguing suggestions on the possibilities of alien chemistries have been provided by Bains [76]. Ward and Benner [77] even elaborated on potential biopolymers. One especially instructive example in this context is a suggestion by Feinberg and Shapiro [78] of how replication could function on an alien world. Instead of coding based on DNA or chemistry at all, it could be based on the alignment of magnets (Figure 1). A similar idea based on non-randomly ordered dust particles has also been modeled [79]. Whether this kind of scheme could actually work, we do not know, but there are no physical or chemical arguments that would indicate this is an impossible scenario.

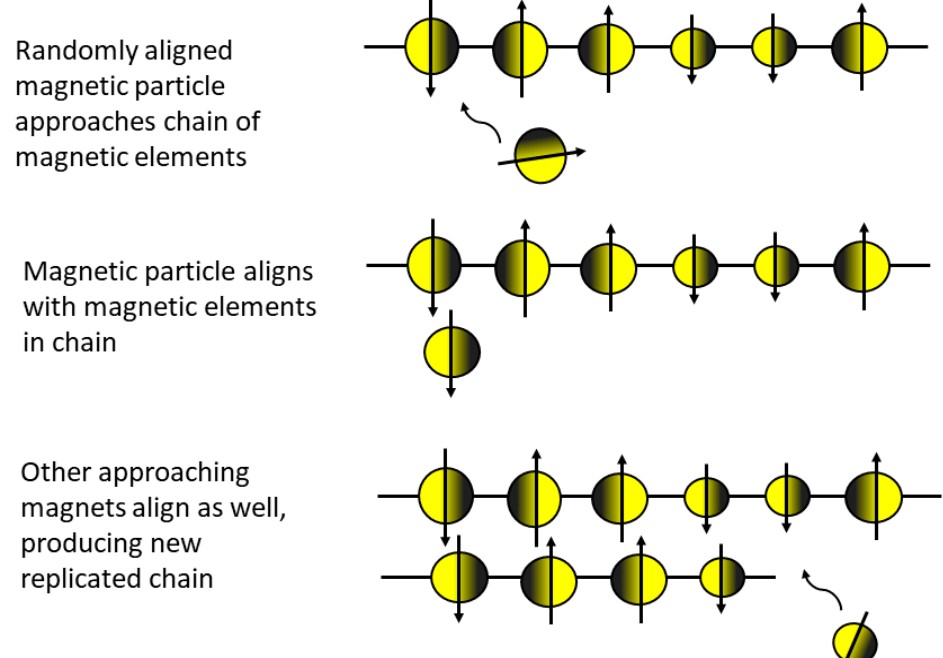

**Figure 1.** Hypothetical replication mechanism of an ordered chain of magnets. Notice that identical components are replicated rather than complimentary and that the chain of magnets is only sensitive to the direction of magnetization, not to magnet size (modified from [7,78]).

*5.2. Unbounded Inorganic Forms*

Growth of inorganic structures in appropriate solvents has long been suggested as a precursor to prebiotic replication. Inorganic templates which could induce formation of organic polymers by complimentary binding of organic building blocks into a defined configuration is a common feature of speculations about the origin of life [80–82]. However, persistence and replication of relatively complex inorganic structures alone has been envisioned and demonstrated to a degree under laboratory conditions.

Complex plasmas are capable of self-organizing themselves into stable interacting helical structures when induced by physical mechanisms involving plasma polarization under appropriate conditions [83]. The resulting helical strings composed of solid microparticles can be topologically and dynamically controlled by plasma fluxes, display metabolic-like reactivity in a thermodynamically open system, and can replicate.

Directional self-organization of soft materials into three-dimensional geometries by the rapid propagation of a folding stimulus along a predetermined path has been engineered in a unique Janus bilayer architecture with chemical and mechanical properties that enable the transformation of surface energy into directional kinetic and elastic energies [84]. This Janus bilayer can respond to pinpoint water stimuli by a rapid, several-centimeters-long self-assembly that is reminiscent of *Mimosa pudica*'s leaflet folding, showing kinetic responses to environmental stimuli.

Self-assembled nanostructures obtained from natural and synthetic amphiphiles have been used to mimic biological membranes [85]. Compounds prepared by coupling tailored hydrophilic and hydrophobic branched segments, when injected into water containing appropriate building blocks, have been shown to generate a rich repertoire of shapes capable of self-assembly.

Colloidal particles that combine mutual attraction, anisotropy, and deformability have been synthesized into self-assembled microcapsules that contain both mutually attractive and repulsive surface groups [86]. Via colloidal bond hybridization, isotropic spheres self-assemble into planar monolayers, whereas anisotropic snowman-shaped particles self-assemble into hollow monolayer microcapsules. Thus, substantial complexity of self-assembled structures can result from modest

changes in the building blocks, providing colloidal equivalents of molecules and micelles. In another combination of nanotechnology and colloidal science, parallel self-assembly of three-dimensional colloidal microconstructs guided by DNA origami were constructed with highly specific geometries that included control over position, dihedral angles, and cluster chirality [87].

Each of these exemplary phenomena embody the three principal elements of living systems enumerated above with some qualification. Most are not encapsulated as within a conventional biological membrane, but all consist of discrete entities with sharp boundaries distinguishing them from their ambient environments. All consist of a higher degree of complexity (lower entropy) than their surroundings, and all convert free energy into kinetic energy or structural elaboration. Each shows a capacity to replicate or at least grow, though the extent to which they can do so autonomously is not yet demonstrated. They all require precisely defined media—in most cases water; though in principle, other solvents might suffice. None of them can be said to display unambiguously all the necessary and sufficient conditions for life, but they serve as indicators of what some very different forms of life, under dramatically different conditions, or originating from a different sequence of events from those that gave rise to life of Earth, could be like elsewhere in the Universe.

### 5.3. Amorphous Organic Forms

Sagan and Saltpeter [88] envisioned hypothetical organisms in the form of "thin gas-filled balloons", perhaps 1 m to 1 km in diameter, that could float in dense atmospheres, deriving energy from photosynthesis or oxidation of methane by producers, which could serve as food for floating heterotrophic consumers. Another speculation [89] proposed vertically floating cigar-shaped organisms, long enough to tap heat energy from lower layers of gas-giant atmospheres (or upper layers of deep aquatic bodies). While a number of reasons make the existence of such organisms unlikely [6], they remain a theoretical possibility for upper layers of dense atmospheres where temperatures allow for the survival or macromolecules.

We have previously suggested the possibility of various forms of amorphous organisms, either attached to an ocean floor or floating through layers of water, harvesting energy from thermal, ionic, or osmotic gradients [6,90]. These organisms are more credible than those in atmospheric habitats because of the rocky planetary bodies (or oceanic bodies with a rocky core) that provide a more plausible substrate for their origin.

### 5.4. Mechanical Forms

The progress of robotics and advances in artificial intelligence require us to consider the possibility that our most likely encounter with alien entities that could have life-like properties will be with machines rather than organic beings. Were an alien explorer to venture into our Solar System and land on the Moon, Mars, or Titan instead of Earth, the closest thing to a living organism they would encounter would be a machine.

As demonstrated by the most sophisticated and artificially intelligent machines that humans have fabricated on Earth, mechanical entities can be endowed with most of the properties of living organisms—their structure is sharply distinct from their surroundings, their entropy is less than that of their environment, and they require energy for their maintenance, information processing, and any work they do. It is only their inability to autonomously fabricate copies of themselves that disqualifies them from a characterization of being alive [7].

The probability that our technology will progress to the point that mass production of intelligent machines will become common place, populating the world with mechanical forms of life that coexist with humans and carry out an increasing number of the activities that humans do, foreshadows the future of intelligent, technologically advanced life on our planet. That somewhere in the Universe, and probably in our own Galaxy [49,91], some technologically advanced forms of life may already have reached that level, is a distinct possibility. On such a world (as on our own eventually), the capability to fabricate intelligent mechanical forms of life will become so sophisticated that the manufacture of such

machines, created initially by intelligent forms of organic life but eventually carried out robotically under the control of an advanced form of artificial intelligence, for all practical purposes can arguably be said to represent autonomous reproduction. At that point, the distinction between organic life and mechanical life will be blurred at best.

As our own behavior has shown, exploration of other worlds by any technologically capable population is likely to proceed initially with mechanical probes. Most exploration subsequently is most likely to be carried out robotically. Therefore, any encounter we have with a technologically advanced society, or any remote detection thereof (on their world by us, or our world by them), is most likely to occur through mechanical forms of life. Whether those entities conform to anyone's definition of "life" will be a semantic issue of little practical importance in deciding how we will interact with them.

## 6. Plausible Evolutionary Trajectories on Other Worlds

The extent to which life essentially as we know it could have followed different trajectories on various types of planetary bodies in our Solar System was described in detail by Irwin and Schulze-Makuch [6]. In this section, we consider how different forms of life, either as we do or do not know them, might have been able to evolve on other worlds, within and beyond our Solar System.

### 6.1. Life in a Subsurface Ocean of an Icy Planet or Moon

Planets or moons with subsurface oceans may be much more common in the Universe than planetary bodies that have a habitable surface environment [92]. Even in our own Solar System there are several worlds that are thought to harbor subsurface oceans, including Europa, Ganymede, Enceladus, Titan, and perhaps even Pluto. In principle, all of these worlds could host life, though it is more likely to be found where active recycling of matter and energy occurs (e.g., Europa, Enceladus) compared to those where this does not occur or occurs only to a minor degree (e.g., Pluto). For example, if the ice layer on top of the subsurface ocean world is convective, oxidants from the surface could be carried to the subsurface liquid oceans to enable oxidative metabolism by organisms in the liquid marine layer [93].

On Earth, a limited biosphere at the hydrothermal vents is sustained by gases released from the rocky mantle, which feed chemoautotrophic microbes. Even multicellular and macroscopic consumers, which feed on these producers, are present. Analogously, those ocean worlds where the rocky mantle is directly in contact with liquid water (e.g., Europa) are more likely to harbor life than those planetary bodies where the liquid water is thought to be sandwiched between two ice layers (e.g., Ganymede). Irwin and Schulze-Makuch [94] modeled the subsurface ocean of Europa with plausible assumptions about the amount of available energy. Their results indicated that Europa could be expected to sustain a modest biosphere, but nevertheless include a complex ecosystem with several trophic levels up to the size of a brine shrimp, occupying a volume approximately equivalent to a residential swimming pool. Other models have predicted both higher [95] and lower [96,97] biomasses on Europa.

The availability of free energy sources in subsurface oceans of ice-covered planetary bodies usually would not include light. This would only be available close to the icy surface, potentially in brine pockets that might be inhabited [98] or perhaps as infrared radiation from hydrothermal vents [99]. However, redox chemistry would still be available as a major energy source and could provide the base of an ecosystem like at Earth's hydrothermal vents. Other possible energy sources include heat, physicochemical gradients like osmolarity, or kinetic energy from ocean currents [100]. Most of those energy sources (except for redox chemistry) yield much less energy than light [7], so any evolutionary trajectory in subsurface oceans would likely be slow to evolve and give rise to entities with little or no motility. Nevertheless, Irwin and Schulze-Makuch [94] showed that a hypothetical Europan ecosystem could be sustained by osmotic and ionic gradients with ionotrophic producers at the ocean floor and osmotrophic producers at the ceiling of the ocean (Figure 2).

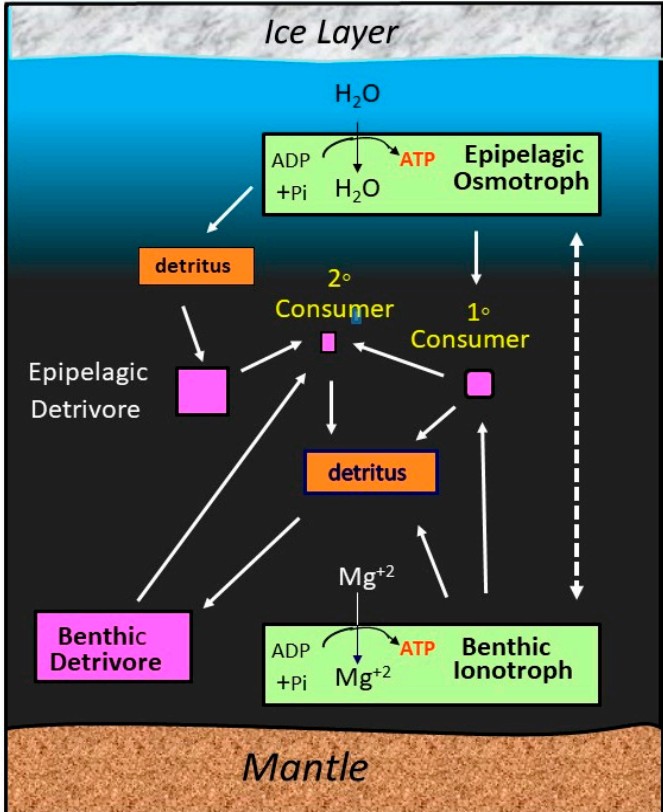

**Figure 2.** Energy flow diagram for a simple hypothetical ecosystem in a subsurface ocean. Producers extract energy from ionic gradients at the ocean floor and osmotic gradients at the ice–water interface. Primary consumers feed on the producers, and secondary consumers feed on the primary consumers. Detrivores at intermediate (pelagic) and bottom (benthic) layers of the ocean harvest energy from the detritus of dead organisms. Arrows show direction of energy flow. Area of rectangles is proportional to estimated biomass of each component [6]. Credit: Art by Louis Irwin, under a Creative Commons Attribution 4.0 International License.

Among the alternative forms of life considered in Section 5, non-photosynthetic microbial mats, either on the ocean floor or the underside of ice sheets at the top of the ocean, seem most plausible, as they were probably among the earliest forms of life on Earth. Amorphous organic forms would appear to be a possibility as well. Thin, spreading films akin to slime molds are plausible. If thin enough, they could be acellular, as they are in the plasmodial form on our planet. Less likely but certainly possible are floating organisms suspended beneath the ice covering, deriving energy from osmotic or ionic gradients, or either attached or floating tubular structures harvesting energy from thermal or saline gradients.

Amorphous conglomerate forms, such as microbialites in various shapes and dimensions, could also appear in such environments. This would suppose evolution to a level of complexity greater than that of microbial mats, which could serve as an intermediate transition to a more complex ecosystem; or it could be a form of life that reaches a constant, steady state, with very slow growth and little change over extended periods of time. Given the low level of available energy in ice-covered oceans, the low environmental temperature, and the stable (unchanging) environment likely extending over geological time scales, evolutionary innovation would be expected to be slow. Notwithstanding these limitations, however, complex ecosystems including macroorganisms could evolve in time.

Thus, subsurface ocean worlds could provide a permanent habitat for life as we do know it, analogous to hydrothermal vent communities on Earth and for alternative forms of life that are rare or absent in Earth's biosphere.

## 6.2. Life on a Barren Planet

The major challenge for life on a desert planet is the lack of surface liquids. Mars is a good example of such a barren planet. It is a frozen desert today, but during a warmer and wetter period [101,102], which may have extended for a billion years as evidenced by lake sediments found at Gale Crater [103], habitable conditions can be inferred to have existed on Mars. However, as Mars became colder and drier, its water became locked up either at its polar ice caps or in the subsurface. Today's Martian atmosphere of approximately 6–8 mbar is so thin that no rain can fall. The only precipitation still possible might be nightly snowstorms or icy microbursts [104] and fog that freezes to the ground [105]. There is also evidence of temporary surface water, which, however, sublimates quickly when it erupts and leaves only a trail of salt [106,107].

Could Earth-like forms of life survive or even thrive in such an environment? Research from deserts on Earth provide examples of how this might be accomplished. Extreme desert environments on Earth include the hyperarid core of the Atacama Desert in Chile and the Dry Valleys of Antarctica. For example, precipitation in the hyperarid core of the Atacama Desert averages about a negligible 2 mm per year. While these environments do not encompass all the stresses that life would experience on Mars, like high radiation intensity and hypobaric surface pressure, they provide good examples of how life adapts to the extreme lack of water.

One strategy is to switch between dormant and active life stages, becoming active during rare rainfall events or when fog provides sufficient moisture. A recent study showed that microbial life became temporarily active after a rain fall event in the hyperarid core of the Atacama Desert [108]. But even without any rainfall, life can survive in this kind of desert. Most deserts are rich in salts, which accumulate due to the evaporation (or sublimation) of water. Some of these salts are hygroscopic, such as the common salt halite, which can attract water directly from the atmosphere. The affinity of the salts for moisture from the atmosphere can be so strong that sufficient water is absorbed to form an aqueous solution. That process is called deliquescence and is used by some microbes, especially cyanobacteria, as a means of survival [109] (Figure 3). In recent lab experiments, archaea survived and metabolized with 100% of water supplied only by deliquescence [110].

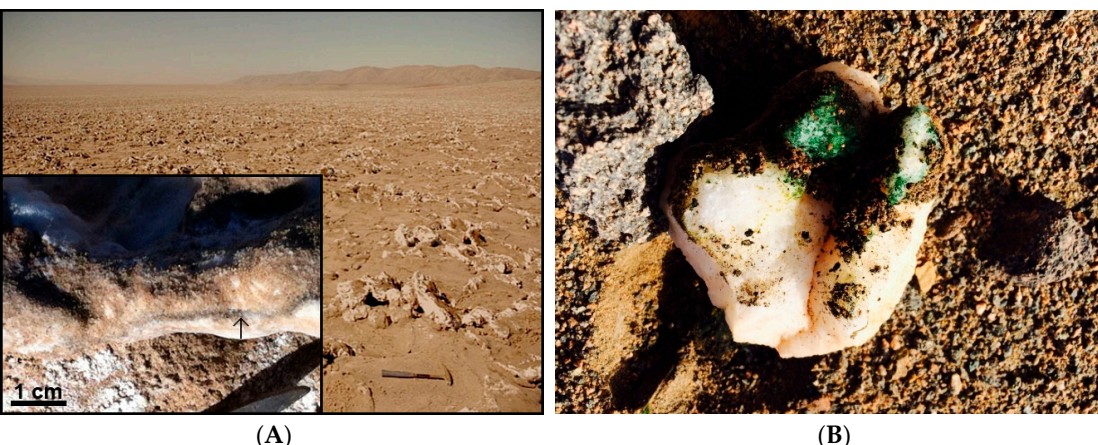

(**A**) (**B**)

**Figure 3.** Endolithic and hypolithic lifestyles in deserts. (**A**) Salt crust in a playa in the hyperarid core of the Atacama Desert. Insert shows a magnification of one salt nodule with a layer of the pigment scytonemin, indicative of microbial colonization. (**B**) A hypolith in the Atacama Desert. The green area are cyanobacteria that live underneath the overturned quartz, sheltered from excess UV radiation and having more access to condensed water. Photography by Dirk Schulze-Makuch.

Another means of survival is the hypolithic lifestyle. In areas where it rains a bit more frequently or fog still occurs occasionally, translucent rocks like quartz are populated on their underside by colonies dominated by cyanobacteria [111]. This location allows them to better access the little moisture

available in the desert and also protects them from excess UV irradiation. These hypoliths are islands of continuous habitability in an otherwise barren desert.

In the most extreme desert locations on Earth, biodiversity is severely limited, with life consisting dominantly or exclusively of autotrophic microbial organisms. There are a few exceptions, such as nematodes, found in dry and cold locations in Antarctica, where they occupy the top of a simple food pyramid with the benefit of oxidative metabolism [112].

Mars may be typical of many planets throughout the Universe that were warmer, wetter, and more geologically active—in short, more Earth-like—in the past. Thus, evolution could have proceeded from pre-cellular chemistry to complex communities of unicellular, microbial organisms, then to multicellular, macrobiological forms. Even if a planet loses its surface liquid and becomes cold and desolate, the fossil remnants of past biota might still be discoverable. This could include remnants of microbialites, as already suggested for Mars [43,47]. If life still exists on Mars today, it would be expected to be microbial and autotrophic, except perhaps in residual environments, such as caves or subsurface liquid reservoirs, where even relatively complex organisms could still exist [113]. Microbial mats or plasmodial biofilms spread along the floor and walls of lava caves where moisture is at least occasionally available are distinct possibilities. In anoxic atmospheres on other planetary bodies lacking alternative robust forms of energy, the size and motility of organic forms of life would be expected to be limited.

In contrast to the Martian scenario, a barren planetary body with no history of having had any liquid is unlikely to have harbored the evolution of any form of life, with the possible exception of unbounded inorganic entities (Section 5.2), and even those would likely have been able to form only in the presence of very special chemical milieus.

*6.3. Life on a Hydrocarbon World*

Planetary bodies, such as Saturn's moon Titan, which are devoid of surface water but densely covered or encased in hydrocarbons, present multiple opportunities for forms of Earth-like life and life that is radically different from life with which we are familiar. Titan is a cold world with an average temperature of about 94 K and an atmospheric pressure of 1.5 bar. The atmosphere is composed mostly of nitrogen with up to 5% methane and other hydrocarbons and noble gases present as trace elements. It is a very reducing environment with $CO_2$ at concentrations in the ppb range and no measurable molecular oxygen. Despite its frigid environment, Titan is a very dynamic world with methane clouds, a hydrological cycle including methane precipitation [114] and hydrocarbon lakes on its surface consisting of a mix of methane, ethane, and some nitrogen.

The availability of copious organic molecules raises the possibility of complex organic chemistry, leading to biochemistry [115,116]. A possible analog of the hydrocarbon lakes on Titan are liquid asphalt lakes on Earth, such as Pitch Lake in Trinidad where deep saltwater mixes with hydrocarbons, somewhat analogous to ammonia–water slurries that mix with hydrocarbon reservoirs on Titan. Water droplets dispersed in the hydrocarbon matrix of such environments are colonized by a microbial ecosystem [117] which transforms the heavier hydrocarbons to lighter hydrocarbons and methane [118]. In the case of Titan, the bottom of the hydrocarbon lakes could be a prime habitat because the high-energy molecule of acetylene produced in the Titan atmosphere is heavier than the ethane–methane mix in the hydrocarbon lakes and would accumulate on the bottom of the lakes. This locality would provide a whole range of microenvironments of hydrocarbons and water in different phases [73]. Associated challenges for life, however, would be the possible lack of silicates and metals to catalyze metabolic reactions.

Thus, in principle, life as we know it could possibly thrive in this type of environment. Earth-type life may also thrive in the hypothesized deeper oceanic layer on Titan thought to consist of ammonia-rich water. Organisms would be most likely heteroautotrophic, given the rich organic substrate on Titan. Some organisms might also be lithoautotrophic, and if enough autotrophic biomass is present, a higher step in the food chain consisting of grazing organisms could be envisioned.

However, Titan also provides a prime example of an environment for life as we do not know it, where, for example, chemical reactions involving metabolisms based on free radicals [73] and polarity-inverted membranes made up of small nitrogen-containing molecules (azotosomes) to interact with the hydrocarbon solvent [119,120], could be used. We have previously outlined possible evolutionary trajectories for the transition from hydrophilic- to hydrocarbon-based environments on planetary bodies such as Titan [6].

*6.4. Life on a Rogue Planet*

Rogue planets are planets that wander through the Universe without orbiting a host star. Based on theoretical considerations, they should be very common, with some estimates claiming that there may be many more rogue planets than orbiting planets in our Galaxy [121]. A high number of rogue planets is also supported by the history of our own Solar System, in which Pluto, Charon, and Triton arguably represent captured rogue planetesimals. Unfortunately, rogue planets are very difficult to detect. The first one detected was a gas giant about six times the mass of Jupiter [122]. Perhaps surprisingly, rogue planets may not be as desolate and barren as generally thought, given that they are moving through space without a star providing warmth and light. There are three types of rogue planets with very different environments. The first type is linked to its possible origin as a failed star, which was not massive enough for nuclear fusion to ignite its core. This so-called sub-brown dwarf would be similar in composition to the gas giants that we know of within our Solar System. Its size could range from several times the mass of Jupiter to as small as Uranus. Life associated with this type of gas giant rogue planet would be extremely unlikely (see Section 5.3).

A second type of rogue planet may not be in orbit around a star but around an active galactic nucleus or black hole, or even a pulsar. Lineweaver, Fenner and Gibson [123] claimed that life would be extremely unlikely near a galactic center, based on the low metallicity and the high-energy radiation outbursts at that proximity to a black hole, though life may be protected from that radiation if it is far enough below the surface. Others [124–126] have argued that these planets may not necessarily be uninhabitable. However, Schnittman [127] suggested that the blueshift of light would make life near a black hole very challenging because incoming light would be amplified due to the relativistic effects to much higher ultraviolet frequencies. Thus, even if rocky planets exist in this region, life may not, because habitability requires much more than the presence of liquid water such as the presence of suitable elements (including carbon and heavy metals), polymeric compounds, a stable atmosphere, a magnetic field, and a recycling mechanism such as plate tectonics.

A third way a rogue planet may originate is from gravitational dynamics during early solar system formation. As planetary bodies come close to very large planets in the chaotic movements before the orbits are stabilized, collisions and near-collisions become more frequent, as known from our own Solar System history. During these encounters, some of the planetary bodies, especially the smaller ones, can be ejected from the solar system and become rogue planets. A planet ejected from a solar system during planetary formation could be very similar to Earth or any of the other inner planets, moons, or asteroids. In fact, life could have originated on a planet before it was ejected from its solar system.

Once ejected, the atmosphere of the terrestrial rogue planet is likely to condense and freeze to the surface of the planet. However, heat from radioactivity in the core could provide sufficient energy to keep the water liquid beneath the icy surface. Abbot and Switzer [128] showed that there can be enough energy to sustain a liquid ocean of water, provided it is insulated by a layer of ice on the outer surface. If so, the environmental conditions would be similar to a subsurface ocean scenario as described in Section 6.1. Light would not be available for life, but life could be based on redox reactions or on alternative energy sources such as thermal gradients [129], osmotic/ionic gradients or kinetic energy [7].

Intriguingly, under certain conditions, an ice layer might not even be necessary. Bada [130] pointed out that some rogue planets could have retained their initial dense hydrogen atmosphere, and water could have condensed and forming oceans. Stevenson [131] calculated that the atmosphere would

have to have at least 100 bars of molecular hydrogen pressure to provide enough insulation to have liquid water on the outer surface of the planet without an ice layer. This could be possible if the planet were to be ejected out of its solar system quickly. Laboratory experiments have demonstrated the viability of *E. coli* and yeast in a substantial hydrogen atmosphere [132]. Light would again not be available as an energy source on a rogue planet with a substantial hydrogen atmosphere, but redox reactions and alternative energies sources could be plausible (Section 6.1).

No single trajectory for the evolution of life on a rogue planet can be predicted, since so much would depend on the starting conditions. If life first emerged as chemoautotrophic in an oceanic light-poor environment, the extent and complexity of the biosphere that could evolve would depend on the availability of the various energy sources proposed in Section 6.1. above. If life on a rogue planet had become dependent on photoautotrophy before the host planet was ejected from orbit around its sustaining star, transition to non-phototrophic energy sources, and therefore the long-term viability of the biosphere on such a planet, would be a challenge. For rogue planets rich in hydrocarbons, the same considerations that apply to Titan (Section 6.3) would be relevant. On planets lacking liquids at any point in their history, the existence of life would be highly doubtful.

### 6.5. Life on a Super-Earth

A Super-Earth is a terrestrial rocky planet with a mass greater than Earth's. Early work suggested that terrestrial planets may exist up to 10 times more massive than Earth, but more recent work has shown that many of the exoplanets with only two Earth masses are already mini-Neptunes rather than rocky planets [133]. Thus, it appears likely that most of the Super-Earths are less than two times the mass of Earth. Heller and Armstrong [134], and more recently Schulze-Makuch et al. [135], suggested that these Super-Earths are candidate planets for being superhabitable—meaning more favorable for habitation than Earth is today.

In principle, any evolutionary trajectory known or imagined having occurred on Earth, including the different forms considered in Section 5, would be possible on a Super-Earth. Conditions even more favorable for the evolution of organic life than on Earth would enable a more rapid pace for evolutionary change, though the rate and extent of evolutionary innovations would depend on the degree of habitat fractionation, the frequency and magnitude of seasonal variations, the frequency with which global changes occurred throughout the planet's history, and other circumstances peculiar to that particular planetary system.

A planet more massive than Earth would have a larger gravitational force exerted on the organisms living on that planet. The effect would be minor for microbial life and organisms that are submerged within bodies of water, but it would have consequences for life on land, especially macroscopic life. Animals and plants would need to invest more energy to grow tall, and more size-subdued forms would be favored compared to a planet with Earth's gravity. For example, organisms with low-lying or prone morphologies (e.g., snake-like animals) would be more common than large, standing animals as on Earth, though stationary plant-like organisms would likely be less restricted due to strong evolutionary pressure to grow toward the light. However, a Super-Earth would probably have a thicker atmosphere due to the stronger gravitational force, making flight a major form of locomotion. Other planetary characteristics, like the vertical extension of the atmosphere, differences in atmospheric dispersal of species and reproductive structures, such as seeds and pollen, differences in redox gradient build up and subsurface void closures would also be affected.

A Super-Earth could be superhabitable if some or all of the following parameters are met [135]: (a) its mean surface temperature is up to about 5 °C higher than on Earth, (b) its atmosphere is moist with 25–30% oxygen and other gases being mostly inert, (c) it is slightly older than Earth, (d) the topographic layout consists mostly of scattered land and water areas with many archipelagos and lots of shallow lakes and seas, (e) the Super-Earth has a large moon at a moderate distance to stabilize the planet's obliquity and generate tidal excursions, and (f) plate tectonics or a similar mechanism allows for efficient geological and geochemical recycling.

Whether life on other worlds inherently requires as long for complex life to evolve as it did on Earth is unknown, but if so, Super-Earths in orbit around spectral type K dwarf stars would have an advantage over planets orbiting G dwarf stars like our Sun, which have relatively short life spans Despite the success of life on Earth in developing a vibrant and complex biosphere, the origin and early evolution of life on the surface of planets orbiting dG stars seems challenging, because young stars of this type emit very intense coronal X-ray and chromospheric FUV (XUV) emissions and high levels of magnetic dynamo-driven activity [136]. Stars with larger mass than our Sun (B-, A-, and F-type dwarfs) have an even shorter life span and, thus, are even less suitable. On the other hand, planets in orbit around low-mass stars, particularly the common dM stars, would have to be much closer to their host star for temperatures to be optimal, and would therefore likely experience high exposure rates of solar wind and high-energy bursts [7], early water loss [137], tidal locking [138], and loss of seasons [139]. Thus, the ideal superhabitable world would be more likely to orbit dK stars, which have a mass that is lower but not too low; a conclusion shared by Lingam and Loeb [140,141] based on a modeling of the erosion of planetary atmospheres due to the UV irradiation and stellar wind.

*6.6. Life on a Tidally Locked Red Dwarf Planet*

If the rotation period of the planet around its own axis becomes equal to its revolution period around its host star, then a planet has become tidally locked and always keeps the same face toward its host star. Even if the rotation and revolution period are initially different, they can get synchronized over time due to the tides on the planet because of the gravitational effect of the star. Tidal locking is a common scenario, especially for planets around M dwarf stars, because their habitable zone (the zone where water can be stably liquid on the planetary surface) is very close to the star. Since more than 75% of all types of stars are M dwarf stars in our Galaxy, many terrestrial planets in the habitable zone will be tidally locked, and so will be some of the planets that orbit larger-mass K dwarf or even G dwarf stars like our Sun [142]. The same can occur to moons around a planet as we observe it with Earth's Moon, which always shows the same side toward Earth.

Initially, it appeared that tidally locked planets would be unfavorable for supporting a biosphere, because it was assumed that the side facing the star would be extremely hot and the opposite side extremely cold, and if life exists at all, it would be located within the narrow twilight zone of the planet. However, opinion started to change more than 20 years ago when a seminal paper [143] showed that tidally locked planets orbiting M dwarf stars can support atmospheres over a range of conditions and could, in principle, be habitable. Tarter et al. [144] agreed that M dwarf stars may indeed be viable hosts for planets on which the origin and evolution of life can occur, because tidally locked rotation was concluded to not necessarily lead to atmospheric collapse, and stellar flaring to not be as harmful to evolution as previously thought. Modeling conducted by Edson et al. [145] confirmed that planets orbiting low-mass stars have areas that are potentially habitable, depending on the substellar point's location relative to the continents, because that largely determines the atmospheric carbon dioxide concentrations and temperatures of the tidally locked planet. Wandel and Gale [146] expanded the range of orbital distances and atmospheric conditions that could support life and suggested that environmental conditions on Red Dwarf planets may even be suitable for the support of oxygenic photosynthesis.

Boutle et al. [147] suggested that atmospheric dust would further increase the habitability potential of tidally locked terrestrial planets by showing that airborne dust would cool the day-side and warm the night-side of such a planet, thus widening the area suitable by life. They also discovered a likely planetary feedback mechanism that would increase atmospheric dust loading and keep water closer to the planetary surface. The net effect would be the delay of water loss at the inner edge of the habitable zone, extending the planet's potential duration of habitability.

While the newer insights gained from modeling show that life could indeed by present on a tidally locked planet [143,145], especially those that have much dust in their atmosphere [147], the chances for the presence of complex life on these planets (and moons) still appears to be low.

Evolution to more complex life is spurred by a highly heterogeneous environment [7] and seasonal variations largely contribute to that heterogeneity. Tidally locked planets have no seasons and no day and night cycles. Earth had all these environmental conditions and it still took about 4 billion years for the first macroscopic, complex forms of life (plants and animals) on our planet to arise. The most significant advance with respect to complexity in the natural history of Earth was the Cambrian Explosion, the rapid diversification of forms and functions after the last Snowball Earth Event. Checlair et al. [148,149] argued that no Snowball Event could occur on a tidally locked planet; instead, it would smoothly transition from partial to complete ice coverage and back. Thus, it seems more likely that the biosphere on a tidally locked planet would remain rather simple, lessening the chances for major transitions in an evolutionary trajectory like the Cambrian Explosion that occurred on Earth.

*6.7. Life in a Planetary Atmosphere*

Given that the biosphere on Earth is so centered on the surface and subsurface of our planet, it may seem strange to consider the planetary atmosphere as a habitat, given its low density, sparse amount of nutrients, and scarcity of liquids. However, even Earth's atmosphere serves as a temporary habitat, mostly for long distance transportation of microbial life. It is now recognized that the transport of microbes from Earth's surface to the clouds is a common phenomenon [150,151] and that clouds harbor a diverse range of microbial life, including archaea, bacteria, eukaryotes, and viruses [152,153]. Viable microbes have even been found in the stratosphere, at an altitude of 38 km [151]. And microbes have been shown to be physiologically active and metabolizing in cloud droplets [153], but so far reproduction has not been demonstrated in the aerial habitat, so Earth's atmosphere cannot yet be considered as a permanent habitat for life. However, if it is not, that should not be surprising, because natural selection has likely focused on temporary survival rather than adopting a life cycle permanently sustained in Earth's clouds given that the environmental conditions on Earth's surface are so well-suited for life [154]. Another challenge for clouds on Earth as a permanent habitat is that they are not continuous, so any microorganism will eventually be deposited back on the surface by precipitation within a few days or weeks at most [155].

However, many other planetary bodies do have permanent clouds such as Venus, Titan, and the gas giants of our Solar System. Venus is especially interesting in this respect because oceans may have existed on the Venusian surface until roughly 700 million years ago [156]. Even if habitable conditions on Venus existed for a much shorter time, life may have been transferred by asteroids from Earth or even Mars when Venus was still habitable; or it may have independently originated on the surface given environmental conditions likely resembling those of the early Earth [157]. At some point in history, the Venusian surface became uninhabitable with temperatures of about 737 K and 92 bar pressure; and life, if it survived, would only have had the atmosphere left as a last refuge. Schulze-Makuch and Irwin [7] summarized why the Venusian atmosphere, particularly the lower cloud layer at an altitude from 48 to 60 km, could be a habitat for life: (1) The lower atmosphere is thick, so microbial transport between the surface and the cloud layer would be easier than in Earth's atmosphere. (2) The clouds of Venus are much larger, providing more continuous and stable environments than clouds on Earth. (3) Current conditions in the lower cloud layer of Venus are relatively benign at 300–350 K, 1 bar pressure, and a pH of 0—conditions of temperature, pressure, and pH under which thermoacidophilic microbes are known to thrive on Earth. (4) Cloud particles are projected to last for several months in the Venusian atmosphere compared to only days on Earth [158]. (5) The Venusian atmosphere is super-rotating, thus cutting the nighttime significantly and thereby allowing for more photosynthesis. (6) Water vapor is reasonably dense in the lower cloud layers of Venus. (7) Oxygenated species, such as $SO_2$ and $O_2$, coexist in thermodynamic disequilibrium with reducing species such as $H_2S$ and $H_2$.

The Venusian atmosphere also harbors an unknown mechanism for absorbing more than half of all the UV irradiation the planet receives. Limaye et al. [159] suggested that this could be the result of

an energy capture process by an aerial biosphere. Schulze-Makuch et al. [160] speculated that this UV absorber could be elemental sulfur, especially cycloocta sulfur ($S_8$), which has the intriguing capacity to adsorb UV radiation and re-radiate it in the visible light spectrum. Thus, microbial life coated with $S_8$ would be able to photosynthesize, in principle. This sulfur-based photosynthesis could produce an ecosystem in the Venusian cloud layer in which the sulfur that is oxidized during photosynthesis is later reduced by chemoautrophic microorganisms.

A related model was suggested by Seager et al. [154], who envisioned hydrophilic filaments in addition to the elemental sulfur that could accumulate the critical liquids the microbes would need. They suggested that the life cycle in the Venusian atmosphere would involve drying out of the microorganisms as liquid droplets containing them evaporated during settling. The smaller desiccated spores would fall into the lower haze layer at an altitude of 33–48 km, from which most of them would eventually return to the lower cloud layer by upward diffusion or convection, where they would be rehydrated by cloud condensation and complete their life cycle. In both hypotheses, some of the microorganism would fall to the Venusian surface and be lost, but microbial reproduction within the lower cloud layer would make up for the lost biomass.

As described above (Section 5.3) a possible aerial biosphere suggested by Sagan and Salpeter [88] consisted of organisms like gas-filled balloons that could exist in the atmosphere of Jupiter or similar planets. However, any kind of aerial biosphere on a gas giant would have the major problem of how life could originate there in the first place. Nearly all origin of life hypotheses require a solid substrate and solid–liquid interactions [7] which do not occur on gas giants (with the possible exception of solid–liquid–gas interactions on microscopic condensation nuclei) at environmental conditions remotely similar to what we consider suitable for life. Thus, most scientists consider the chances for life within gas giants to be practically zero. However, there may be gas giants outside of our Solar System with very different environmental conditions, which should not be automatically excluded as possible habitats for life. As the example with Venus shows, the natural history of any planet may be critical for evaluating whether life might possibly be present in a planetary atmosphere.

## 7. Implications for Fermi Paradox

The ever-rising number of confirmed exoplanets points to a staggering total of other potential habitats for life in our Galaxy alone. While the vast majority appear to be unsuitable for life as we know it, the sheer number of possibilities argues strongly for the occurrence of some forms of life on some other worlds. As this review has pointed out, life need not be restricted to a "habitable zone" narrowly defined by characteristics suitable for life as it occurs on Earth. Therefore, when all conceivable forms of life are considered, the number of worlds on which life has possibly evolved to some degree of complexity is great; and quantitative models based on objective criteria bear this out, with estimates of mature biospheres in our Galaxy ranging from 0.5 million [161] to 100 million [49]. A logical question follows: what are the chances that we will ever come in contact with any alien form of life?

The possibility that large numbers of planets bearing intelligent life exist, yet we have no evidence of them, is named the Fermi Paradox after Enrico Fermi, who allegedly posed the dilemma over 70 years ago. An alternative term, the Great Silence, has been suggested, since Fermi never published his anecdotal query, and others had raised it prior to him [162]. Numerous explanations for the so-called paradox have been offered [163], a few of which are more credible than most. We have consistently argued that the absence of contact with alien forms of life technologically capable of making themselves known to us is largely a matter of statistical improbability, given the immense distances from Earth to the nearest plausible home for such life, the tiny target that Earth represents on a cosmic scale, the likely disparity in time of overlap between our mutually capable technologies and the extremely narrow window on a geological time scale of our ability to receive and understand a relevant signal [7,49,90]. In light of the information conveyed in this review, we will briefly reconsider its implications for the Fermi Paradox.

The argument has been made that, even though simple life may be common in the Universe, complex life is truly rare. The best-known version of this argument is the Rare Earth hypothesis [164] which attributes the rise of complex life on Earth to the remarkable number of narrowly constrained circumstances that enabled our planet to sustain evolution for an extended period of time in diverse habitats through recurrent environmental changes. Since environmental conditions on the vast majority of planetary bodies lie outside the tolerance for most complex forms of life on Earth, the only life on those worlds is assumed to be microbial. The Cosmic Zoo hypothesis [56] argues, on the contrary, that once life originates, its evolution can take many paths to complexity in all sorts of environments; therefore, complex life will eventually arise if the planet in question stays habitable for a sufficiently long time. Thus, though complex life will certainly be rarer than microbial life, it still could be relatively common. However, there are two limitations to the Cosmic Zoo hypothesis. First, since it is unclear how life originated on our planet, it is possible (although unlikely) that the origin of life is itself such an improbable event that it very rarely occurs. Secondly, the Cosmic Zoo hypothesis also provides no clear answer to the likelihood of the emergence of technological intelligence, given that it has arisen only once (with us) in our entire planetary history. Thus, it too could be a highly improbable event. Both models, though for different reasons, lead to the assumption that technological intelligence is very rare. Even if it does emerge, its self-destruction in a relatively short time on a geological time scale is a high probability [165]. These arguments collectively do not rule out the existence of complex life on other worlds but do point to the statistical improbability of contact between two mutually capable technologies originating independently.

The one factor that conceivably could make contact slightly more probable would be the evolution of mechanical life. The early stages of that trend are already underway on Earth, and the reliance on machines for space exploration by humans now attests to the likelihood that space probes from any technologically capable biosphere are likely to be primarily if not exclusively robotic—especially in the earliest phases of that exploration. Since the reach of and acceptable risk for robotic exploration is much greater than for that of human exploration, it will remain at the forefront of exploratory strategies by our species. Whether organic forms of life on other worlds would be similarly restricted is unknown, but the advantages of robotic exploration would presumably apply on those worlds as well.

Assuming that the outreach into space of any technologically capable beings anywhere will likely be more extensive if conducted robotically, the chance that our machines will come in contact with theirs is marginally enhanced; but the statistical improbability of even that remains daunting. If that is indeed the case, the lack of evidence for intelligent life on other worlds should be viewed less as a paradox and more as an inevitability.

## 8. Ongoing Studies and Prospects for Further Investigation

While the likelihood of contact with intelligent forms of life beyond our Solar System is very low, the prospect of finding some alien forms of life, if they exist, within our Solar System is reasonably good. Beyond the potential discovery of living organisms (or evidence of their past existence) on other worlds, laboratory simulations, theoretical models, and study of analog habitats on Earth can further our understanding of what the range of possibilities is for life both as we do and do not know it.

Table 1 lists the forms of life that theoretically could exist under different planetary conditions and the possible ways they could be observed and studied. The table documents the fact that a range of planetary conditions, capable of hosting both familiar and unfamiliar forms of life, are accessible to astrobiological investigation at our current state of technology.

**Table 1.** Habitats and forms of life plausible on other worlds with supporting observational and experimental possibilities.

| Habitat | Solar System Examples | Life Forms Possible [1] | Observational Possibilities with Current Technologies | Experimental Possibilities |
|---|---|---|---|---|
| Rocky/water surface with dense $N_2/O_2$ atmosphere | Earth | Micro/macro individuals Amorphous inorganic Amorphous organic | Direct, current, ongoing | Direct, current, ongoing |
| "Super Earth" (larger and warmer than current Earth) | None | Micro/macro individuals Amorphous inorganic Amorphous organic | Remote Analog habitats on Earth | Lab simulations Studies of analogs |
| Barren rocky, with previous water, no or little atmosphere | Mars | Micro/macro individuals Amorphous inorganic Amorphous organic | Robotic Human exploration Analog habitats on Earth | Analysis of return samples In situ instrument analysis Lab simulations Studies of analogs |
| Barren rocky, no previous water, with little or no atmosphere | Mercury | Amorphous inorganic | Robotic | Lab simulations |
| Ice-covered global ocean | Europa Ceres (?) Enceladus | Micro/macro individuals Amorphous inorganic Amorphous organic | Analog habitats on Earth Robotic | Studies of analogs In situ instrument analysis Lab simulations |
| Rocky/hydrocarbon surface with dense $N_2/CH_4$ atmosphere | Titan | Micro/macro individuals with exotic biochemistry Amorphous inorganic Amorphous organic | Robotic | In situ instrument analysis Lab simulations Studies of analogs |
| Rogue planet with various surfaces and atmospheres | Triton (ancestrally) | Dependent on surface, atmosphere, and planetary history | Very remote | Lab simulations |
| Tidally locked with planet or Red Dwarf | Moon, Io | Dependent on surface, atmosphere, and history | Direct human exploration of Moon Robotic on Io Remote for exoplanets | Analysis of sample return from Moon Study of analog habitats for Io Lab simulations |
| Gas giant or rocky, with dense atmosphere | Gas giants, Venus | Microbial Amorphous gaseous | Robotic Spectral signatures | Robotic probe of gas giants Sample return from clouds of Venus |

[1] Mechanical forms of life originating in other solar systems are possible in all habitats.

The cold, barren planet of Mars is close enough to Earth to make return of samples feasible. Just such a mission is currently underway with the prospect of returning physical samples to Earth for analysis [166]. Human missions to Mars are anticipated by mid-century which will enable direct observation of microbial and possibly macrobiological samples in sequestered environments. In situ evidence for microbial life on Mars remains controversial [7,108], but newer and more capable instruments than the Viking landers, either on their way or in development for delivery to Mars, such as the ExoMars Lander [167], demonstrate the potential for robotic collection of relevant data. Numerous studies of analog environments on Earth likewise should continue to shed light on possible forms of life on a barren planet like Mars [108,110,168,169].

Venus likewise is reachable from Earth within a reasonable period of time, and the Stardust mission which returned physical samples from a comet in 2006 suggested the feasibility of a sample return mission from the atmosphere of another world [170]. While no sample return missions are currently being planned, they have been proposed [171]. Spectral signatures of possible biochemical origin in the clouds of Venus also suggest a remote detection strategy [159]. Robotic probes from Earth have entered the atmospheres of both Jupiter and Saturn, returning valuable data on the physicochemical characteristics of their clouds.

Ceres, Europa, and Enceladus, among others, provide examples of ice-covered oceanic worlds reachable by robotic probes from Earth with current technology. Several habitats on Earth have been

proposed as analogs for this type of planetary habitat [172–175]. Experimental approaches for detecting chemical biosignatures, either in the lab [176] or by slow-moving fly-by probes [177], have been proposed.

While no Earth conditions are known to be comparable to the frigid hydrocarbon atmosphere or petrochemical lakes of Titan, an active community of archaea and bacteria was found to inhabit the liquid hydrocarbon matrix of Pitch Lake in Trinidad and Tobago [118]. The possibility of methanogenic microbes in hydrocarbon environments like those on Titan have been modelled [178], and analogs of the tholins in Titan's atmosphere have been produced in the laboratory by irradiating gas and ice mixtures with radiation simulating solar ultraviolet (UV) photons and the charged particles trapped in Saturn's magnetosphere [179]. The highly successful 13 year Cassini–Huygens mission to Saturn and its moons has provided detailed data on Saturn's atmosphere and its numerous ice-covered satellites (most notably, Enceladus); and the Huygens lander on Titan—the only successful landing of a probe on a planetary body in the outer Solar System—revealed invaluable information about that hydrocarbon world [180].

Triton is even colder than Titan, and laboratory simulations of ices that could occur there, as well as on Pluto and Charon, have been the subject of organic chemistry experiments [181]. These planetary bodies may represent formerly rogue planets, captured by Neptune [182] and the Sun, respectively.

Samples returned from the Moon have thus far revealed no evidence of current or past life, but the discovery of limited stores of water on the Moon [183] suggest it is not impossible that the Moon might have been temporarily habitable about 3.5 billion years ago when most of the basalt mare formed and the Moon might have had an atmosphere of 10 mbar [184]. The return of humans to our satellite will provide an opportunity for further sampling of the nearest case of a tidally locked body we have. Mercury would probably provide a more instructive example, but no planning for such a mission is currently underway to our knowledge.

Missions to exoplanets are not feasible at this time, but remote sensing opportunities continue to increase, and more powerful options are anticipated in the future. Currently most exoplanets are detected either by the radial velocity method, transit photometry, or astrometry, but the main problem remaining is that we cannot distinguish whether the light photons come from the star or the planet. One option for the future is to use interferometry to eliminate photons from the star, by either using a coronagraph within the telescope or by blocking photons from a star with an occulter such as a starshade [185]. The next stage in exoplanet research is to gain information on the environmental conditions of exoplanets, including the major gases present in the atmosphere, and whether the planet has oceans and continents, polar ice caps, and clouds. With the missions of the next generation such as the James Webb Space Telescope (JWST), the PLAnetary Transits and Oscillations of stars (PLATO) mission, the Nancy Grace Roman Space Telescope (NGRST), the Transiting Exoplanet Surveying Satellite (TESS), the New Worlds Mission (NWM), and the ground-based European-Extremely Large Telescope (E-ELT), this should be possible in principle [186]. For example, an ocean could be detected analogous to how the polar lakes on Titan were imaged by the Cassini orbiter: through the glint of sunlight reflected off the liquid areas [187]. An especially ambitious idea of a future remote sensing mission is using the Sun as focal lens for gravitational microlensing [188], which would allow to image the exoplanet not only as a single pixel, but with a resolution in the range of $1000 \times 1000$ pixels. However, the disadvantage would be that the observing telescope would have to be at a distance of at least 550 AU from the Sun in interstellar space, the telescope would have to be exactly aligned, and follow-up observations would not be possible due to the rare alignment [7]. Nevertheless, this shows that creative ideas are available and that, in time, our remote sensing abilities should tremendously improve even if site visits are not possible for the foreseeable future.

## 9. Discussion and Conclusions

We began this review with the argument that, contrary to some views, reasonable consensus on a theory of life does exist in the life sciences. That theory is that life is a physical form of

matter encapsulated or otherwise distinct from its environment maintaining a lower entropy than its surroundings, using energy from the environment for internal maintenance and activity, and capable of autonomous reproduction. Such a theory encompasses all the forms of life on Earth that can be considered unambiguously to be alive, including some exceptional examples, like the amorphous myxobacteria ("slime molds") and highly extended clonal organisms like coral reefs and certain soil mushrooms. Beyond these forms of life as we know them, however, other entities that could satisfy the characteristics of life as defined by the theory above but as yet unknown to us can be envisioned including organic forms with exotic biochemistries, dynamic inorganic matter, and self-replicating machines.

The probability that any given form of life, either known or unknown to us, exists on other worlds depends on the planetary history of that world, the nature of its environment, and the suitability of given forms of life to the historical and environmental constraints of that world. In agreement with most astrobiologists, we assume that while rare, life occurs redundantly throughout the Universe but overwhelmingly in microbial form. This is because most planetary bodies harbor environmental conditions that are extreme by terran standards; and on our planet, only microbes are capable of surviving in the broadest range of environmental conditions, are the simplest and most ancient forms of life, and have persisted on Earth relatively unchanged from their early stages of evolution. The sheer number of planets throughout the Cosmos, however, implies that life on some of them must have evolved to a size and level of complexity beyond that of microbes as we know them on Earth.

Our own Solar System features a good sample of the different planetary conditions available for potential habitability. A mineral-rich ocean beneath a cover of ice as on Europa, and the interface between a rocky core and the water below, could serve as a viable habitat for the origin and evolution of life to a considerable degree of complexity, provided an alternative to photosynthesis from light in the visible spectrum is available. On a cold, bleak, anoxic world like Mars, remnants of a more complex evolutionary trajectory initiated when that planet was warmer, wetter, and more Earth-like could result in the continued survival of microbes in underground water reservoirs and even more advanced forms in sequestered habitats like lava tube caves. Endolithic microbes could still persist, as well as minute organisms able to cycle from dormant, desiccated to active, hydrated forms when occasional moisture makes such a transition possible. The ready availability of reduced carbon compounds on planetary bodies like Titan provides an abundance of building blocks for the organic chemistry with which we are familiar, but the extremely frigid temperatures and the hydrophobic liquids that such temperatures mandate at or near the surface would entail a form of life dependent on highly exotic biochemical systems.

Rogue planets ejected from their solar system of birth could retain a biosphere begun under earlier, more propitious conditions, provided environmental conditions and the availability of a non-phototrophic energy source permit. Planets tidally locked to their central star would be a challenge for the evolution and persistence of life, once synchronous rotation had been reached. Both tidally locked and rogue planets could harbor life in forms totally unknown to us. Super-Earths are the planetary form most likely to contain life as we know it, perhaps to a more advanced extent.

Venus and the gas giants present environments almost certainly too harsh for the survival of life as we know it, except for the lower clouds of Venus which could still harbor microbial life evolved from a surface origin before the surface became too hot for the preservation of macromolecular integrity. The upper cloud layers of the gas giants conceivable could support floating life forms, and deeper into their atmospheres, exotic forms of life are conceivable, except that their origin under such conditions is almost impossible to imagine.

Until we have examples of life from another world, we cannot say whether life as we know it is the only form of life possible. Thus, restricting our consideration to a "habitable zone" predicated only on life as we know it is premature. Finding any other form of life would significantly increase our confidence in the range of possibilities for the nature of life on other worlds. Sample return missions from Mars are already underway, and human exploration of that planet is planned within two decades.

Robotic exploration of the ice-covered worlds, Ceres and Europa, and a sample return from the clouds of Venus, are technologically feasible now; plus, a robotic return to Titan is in the planning stages. Therefore, light could be shed on the range of possibilities for life on other worlds within the lifetimes of most current astrobiologists.

**Author Contributions:** Both authors contributed equally to the conceptualization, writing, and review of the manuscript. All authors have read and agreed to the published version of the manuscript.

**Funding:** Part of this Research was funded by the European Research Council Advanced Grant "Habitability of Martian Environments" (HOME, no. 339231).

**Acknowledgments:** None in addition to the funding.

**Conflicts of Interest:** The authors declare no conflict of interest.

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
