# Peer review of "The Astrobiology of Alien Worlds: Known and Unknown Forms of Life"

_universe, doi:10.3390/universe6090130_

Round 1

Reviewer 1 Report

The manuscript is timely, well organized and well written. I enjoyed reading it.

Below, there is a remark whose implementation will make the manuscript publishable.

Section 6 should be enriched by a further subsection devoted to an intriguing scenario which very recently gained attention: life on Earth-like, starless planets orbiting supermassive black holes whose accretion disk would provide them with the necessary amount of radiation. It has been recently shown that such planets may, actually, form very numerously, and that even certain effects of General Relativity may have, among others, impact on their sustainability of complex forms of life/civilizations.

To this aim, cite and discuss the following references

Bakala et al., The Astrophysical Journal, Volume 889, Issue 1, id.41, 2020

Manasvi, The Astrophysical Journal, Volume 877, Issue 1, article id. 62, 8 pp. (2019)

Iorio, The Astrophysical Journal, 889:152 (6pp), 2020

Iorio, Astrophys. J. 896 (2020) 82

J.D. Schnittman,  arXiv:1910.00940, 2019

Wada, K., Tsukamoto, Y., & Kokubo, E. 2019, ApJ, 886, 207

Author Response

REVIEWS AND RESPONSES TO Universe PAPER

Responses are in italics and bold

Reviewer 1:

The manuscript is timely, well organized and well written. I enjoyed reading it.

Below, there is a remark whose implementation will make the manuscript publishable.

Section 6 should be enriched by a further subsection devoted to an intriguing scenario which very recently gained attention: life on Earth-like, starless planets orbiting supermassive black holes whose accretion disk would provide them with the necessary amount of radiation. It has been recently shown that such planets may, actually, form very numerously, and that even certain effects of General Relativity may have, among others, impact on their sustainability of complex forms of life/civilizations.
To this aim, cite and discuss the following references

Bakala et al., The Astrophysical Journal, Volume 889, Issue 1, id.41, 2020

Manasvi, The Astrophysical Journal, Volume 877, Issue 1, article id. 62, 8 pp. (2019)

Iorio, The Astrophysical Journal, 889:152 (6pp), 2020

Iorio, Astrophys. J. 896 (2020) 82

J.D. Schnittman,  arXiv:1910.00940, 2019

Wada, K., Tsukamoto, Y., & Kokubo, E. 2019, ApJ, 886, 207

A new paragraph with the above references has been added to Section 6 to cover this scenario.

Reviewer 2 Report

General aspects

First of all, let me congratulate to the authors to develop a manuscript in such a difficult topic, where the balance between theoretical - hypothetical argumentation and realistic, established scientific methods are diffiucult to find. As a results, I suppose they are expected to have strong critics from the referees – thus please consider my critical points as supportive and constructive comments.

The topic of the manuscript is important: the possiblity to identify some methodologial aspects what are like and how to search for the possilibity of life that is highly different from those we know on the Earth. Although it is quite difficult to approach such a theretical topic with a clear and well structured format, the manuscript could and should be improved further with these aims. The language of the manuscript is ideal, the structure is moderately good, however more information including figures or talbes that explain the possibilites and support their comparison would be useful. The referee suggests major revision and encourages the authors to work on the manuscript as it has potential and the topic is poorly discussed n the literature. Some general suggestions are listed in the general aspects below, however please consider carefully the specific suggestions too, as there are few (not many) quite important one also among them. An imprtant

General suggestions to improve the manuscript:

  • Although the citations cover a wide and representative part of the corresponding literature, some more and specific citations are necessary to improve the context.
    • And it is important in such a topic to indicate already at the beginning that astrobiology is doing a wide range of research in this domain even these are targeting the possiblit and detectability of Earth-like life (and cite: https://ui.adsabs.harvard.edu/abs/2016AsBio..16..561D/abstract)
    • At the definition of life (line 59) please cite Ganti’s pioneering work, what was prouced early (Ganti, T. (1971). The Principle of Life (In Hungarian language). Budapest: Gondolat.) but became widely known only later: Ganti, T. (2003). The Principles of Life. Oxford University Press or Ganti, T. (2004). Chemoton Theory Vol I and II. Kluwer.
  • A possible improtant issue is the „ability for evolution” and its relation to reproduction. This is a theoretical question: if there is a potential life form that fulfil the main requrements listed in line 93 and later, but if the potential organism (or potential form of life) is able to reproduce but not able for mutation (or similar process), it will probably not evolve toward „higher level”, and its classification as life is questionable. This aspect is partly touched in the lines 94-100 – however this might requre more discussion there.
  • It would be better to have a bit more discussion on LUCA in the point of view how its estimated characteritics might widen/sharpen our definition of life and possibltiy to identify alien life forms. It might improved any model apporach of alien life.
  • section on Fermi paradox: this is a good intellectual adventure, however it would be much better to put less emphasis on Fermi paradox (and shorten this part) as there are more topics to be developed, which are closer related to this work (see the other general comments)
  • 1080-1088 lines: this part is too much Solar System oriented. Please consider to expand to a wider outlook (as this part is the Discussion and Conclusion section) to extrapolate to exoplanets and exomoons
  • It would be useful and almost necessary to provide some further conclusive aspects at the end of the work. In the current form it is an interesting collection of exotic possiblites, but does not provide specific findings. Althoug it is quite difficult to provide some specific conclusions of an almost theoretical work, but please consider, could the authors give further and useful conclusive remarks in the following aspect:
    • What type of further research topics could be focused on, based on the listed findings and suggestions (even if the conclusion is not a realistic plan for how simulation of alien life properties could be made, but this should be also noted)
    • It would be interesting to see some theoretical „flow down” of arguments, what the discussed topics in the work suggest, regarding how to approach the problem, and is there observational possiblity? How widen the observed characteristics or their interpretation to make the observations based research more relevant for not „Earth-like life”?
    • Would it be possible to make a table with the topic groups listed in 3.1, 3.2, 3.3, 5.1, 5.2., 5.3 as theoretical possiblities in the lines, and columns on how to approach the given possiblity, could it be tested in laboratory or with computers, doesthe gven lie form has relevant consequences on its environment with changing it, is there some theoretical observatonal possiblitiy etc.?
    • For example based on the text between 481-497 lines, would it be possible to say something specific for observability (not about specific instrumentation but about what type of approach might be useful for the detection of such life forms)?
    • It might be also a possiblity to compare the listed hypothetic types of life according to their estimated ability to change their environment.

Specific aspects

39 line „then life on other worlds is likely to be very rare”

Do you mean life like on Earth? (based on water and organics)

62-66 lines: it is good to list many requirements, even if different groups are used in different approaches. One what I miss is to specifically mention of „such coding mechanism that provides heritage” – as the „reproduction” might not necessarily cover this.

64 line: please discuss briefly (even with some words in bracket), why homochirality is part of the definition of life and not a requirement / prerequisite for the origin of life (for chemical based life). These two aspects are close to each other, but not the same, thus would be good to mention, especially if you compare homochirality to the 2, 4, 9 items in your list.

66 line „teleology” is listed as one of the definitions, however there is no any reason mentioned for that, and the referee thinks it should not fit to this list of definitions. Please provide reason for that or delete. Even if this comes from a cited work, please discuss briefly why is it listed – many readars will lack (can not see) the reason to include this.

80-82 lines: please define what do you mean on „alien life” and on „Earth-like life”

96-97 lines: „First, it can only apply to multiple generations…” I’m not sure in this. Processes like innovation in biological systems might happen in other ways than generations, in theory a system could improve itself without new generations. I suggest to consider and discuss this with more details. If you take a system that is considered to be „life”, it should have reached this state from an abiotic state in some way. If you know other way than the evolution (in a wide sense) than please discuss this. But the contradiction (no evolution is necessary for all types of hypothetic life, but in this case how could emerge) should be solved. And the second part of the argumentation („it implies change by natural selection, which even in some terran forms occurs barely at all over extended periods, and in alternative (e.g. mechanical) forms of life might not occur at all.”) please formulate the sentence better than the current version.

112-113 lines at the end of the Introduction section: it would be useful to put one sentence on how do you discuss the problem area in the rest of the work and what is the logical structure behind.

120 line: „recurrent heavy” please define better, what do you mean? Early bombardment around the end of primary accretion plus LHB together? Or stochastic impacts or what?

187 line: is there any summary, consequence or other final outcome that comes from this section? It ends a bit „abruply”.

193-197 lines: interesting part, is there any theroetical consequence on the observabilit of such life forms?

201-203 lines: interesting! would be good to have a bit more info here on this

208-208 lines: also more info would be useful

232 line: at „…planetary bodies, as already claimed for Mars [39, 42, 43].” please also cite this, relevant for next mission: https://ui.adsabs.harvard.edu/abs/2016OLEB...46..435K/abstract

240-242 lines: a bit more information would be useful here.

256-256 lines: „While structures like coral reefs would be easier to detect than other forms of life on other worlds because of their size” OK, but indicate if the observer has very high spatal resolution. Current attempts are more targeting specific spectral aspects than simply large scale features on a planetary surface. Your finding is relevant for the far future optical interferometers.

270-272 lines: suggest to consider the so-called infra-biolgical systems (Fernando and Szathmáry 2009) also

277 line: „Most of life on Earth has remained in its ancestral form, evolving little since its origin.” Absolutely not clear, are you sure did you intend to wirte this?

281 line: „requires preconditions that take time to develop” please give example

284-285 „create a complex new function” please give example, and consider here to cite: Szathmary, E. and Maynard-Smith, J. (1997). From replicators to reproducers: the first major transitions leading to life. Journal of Theoretical Biology, 187:555–571.

293 line „relatively soon,” please indicate at least the scale of duration

290-298 lines: please indicate is this text for surface or subsurface (or both) biospheres? And for chemo- or for photosynthesis?

329 line: endosymbiosis should be a bit more detailed, especially regarding tis potential under „different” life forms or its role on the evolution of Earth-like life. Is there some furher theoretical possiblity, what could have been exploited by endosymbiosos but did not happen on Earth? This is a very difficult question I see, please skip if you have no relevant idea for the revised manuscript.

353 line: „can occur in very little time” please quantify, at least the scale of duration

358-362 line: please mention briefly the relevance of this on biomorphological structures

267 line: „oxyen” – it would fit well to the topic of your manuscript to indicate, what other „easy” and potentially accessible energy source could be exploited by chemical oriented life forms

361-372 lines: these arguments show that the effects emerged not as „external forces” but as an „internal mechanisms” by the interaction of different organisms – this might be mentioned, and also discussed somewhere that the components of the biophere migh exaggerate, spin up its own impovement and evolution

384-385 lines: „radically different environments from those in which terran life evolved” please present / formulate this differently, to make it more understandable

417 line: „Life as we don´t know it may” is the „it” necessary here?

around the end of 5.1 section, would it be interesting to consider the followign possiblity? https://www.newscientist.com/article/dn12466-could-alien-life-exist-in-the-form-of-dna-shaped-dust/

356-360 lines (about there but all along the 5.4 section): do you consider possibility for natural emergence of such hypothetic machines? Or only by already evolved complex life (like those on Erth as technical civilisation)? Please indicte, this is not evint from the curent text.

575-584 lines: please cite as a discussion of various factors on habitability this work: https://ui.adsabs.harvard.edu/abs/2016OLEB...46..473K/abstract

Figure 2: why geothermal heat is not indicated?

633-634 lines: „the low level of available energy in ice-covered oceans” have you considered geothermal heat? In ideal case you should cite the corresponding reference or present some numerical comparison/argumentation why such subice oceans are „energy poor”? And also in what aspect? Relatively to the total habitable volume, compared to what?

646-647 lines: „However, as Mars became colder and drier, its water became locked up either at its polar ice caps or in the subsurface” please cite: https://ui.adsabs.harvard.edu/abs/2012AsBio..12..586K/abstract

649-652 lines: „There is also evidence of temporary surface water, which, however, sublimates quickly when it erupts and leaves only a trail of salt [95].” please consider deliquescence already here: https://ui.adsabs.harvard.edu/abs/2015NatGe...8..357M/abstract and https://ui.adsabs.harvard.edu/abs/2020Icar..34013639P/abstract

682 line: „their underside by colonies dominated by cyanobacteria” please cite https://ui.adsabs.harvard.edu/abs/2002IJAsB...1..305C/abstract

703-706 lines: „a barren planetary body with no history of having had surface liquid is unlikely to have harbored the evolution of any form of life” do you mean subsurface habitability does not matter much here?

771 line: „Switzer (, 2011 #2165} showed” please provide proper citation

817-825 lines: this part contains some poor text nd content, what should be develped further toward higher level, more sophisticated and more scientific parts. Arguing „For example, snake-type organisms” is not relevant, however a bit more established content might improve this part substantially. I would suggest focus on more general aspects and what is relevant for simpler organism (as the approach of the manuscript is to consider different possbile life forms, it should be broader than such very Earth-relevant specific items than „snake type”). Please consider the effect of stronger gravity on the verical characteistics of the atmosphere, on global nutrient migration, difference in redox gradient build up compared to lower gravity, restrictions in material circulation, subsurface void closure depth etc.

830 line: „the Super-Earth has a large moon at a moderate distance,” please explain why, suggest to consider that larger mass means larger inertia and larger stability of rotational parameters without a moon

839-840 lines: OK, but consider these are relevant only for surface life

861 line: „cannot host a biosphere” consider to modify to „unfavourable to host…” and cite the corresponding work

863 line: „and if life exists at all it” do you mean „surface life”?

around this above mentioned part it would be useful to discuss, could incrased irradiation (wether UV or ionizing radiation) support the build up of redox gradient, and suggest to cite and use this work: https://ui.adsabs.harvard.edu/abs/2018PrPNP.102....1L/abstract

884-889 lines (and around this part): it would be useful to discuss more the role of catastrophes, which can have favourable and unfavourable consequences for a biosphere

887-888 lines: „Evolution to more complex life is spurred by a highly heterogeneous environment [3] and seasonal variations largely contribute to that heterogeneity” OK, but please consider the abundance of available nutrients and energy sources, what do metter also

889-890 lines: „most of them will also not be” consider to modify to „most of them might not be” and discuss briefly why

902 line: „it may seem strange to consider the planetary atmosphere as a habitat” it would be good to provide some furhter arguments why it can be considered as a poor habitat, or at least why less favurable compared to others

919 line: „Venusian surface until about 700” consider to modify to „roughly about” as this 700 Ma is really a very rough estimation

922 line „given the similar environmental” consider to modify to „possibly resmble” instead of similar

934-947 lines and around here in this work: please try fo put more emphasis on exoplanets (and less is enough from Venus then currently discussed)

963 line: „which do not occur on gas giants” maybe, but it could be mentioned that this environment might be also rich in solid-liquid-gas interations in microscopic scales if there are many condensation nuclei and strong vertical mixing

1035-1039 lines: this is a hypothetical argumentation, to state this, numercal calculations would be necessary – what goes beyond of the scope of this work. I suppose it should be mentioned that this paragraph is based on „simple theoretical argumentation”

1061 line „self-replicating machines” and earlier related parts: please emphasize that the origin of such machines requires other forms of life to produce them (even if they might be self standing later), what could be taken as a unique aspects among the other forms of life, which are all have natural origin. It would be a too far reaching discussion to discuss the difference between naturally originated life and „artificially produced” life- however would be useful to note this.

1082 line: „even more advanced forms in sequestered habitats like lava tube caves” please note that lava caves on Mars might be relatively small (compared to other habitats) and need not be connectedor fewmight be connected, what might not be „enough” for the emergence of advanced living organims

1096 line: „gas giants present environments almost certainly too harsh” the referee still not feel that this statement became well established in the earlier parts of the manuscript. Could you consider the wide range of gas giants might be present among exoplanets? As they cool while getting oder, range of condensation can happen inside them, what together with various mixing processes might make interesting environments for the origin of life – or might not make. There also migth be liquid surfaces in the interior. But to state they are harsh is not established or proven in this manuscript.

1105 line „it is premature.” yes, but this is required to plan instruments and run research project. Please indicate this also.

After 1112 line: I miss here more discussion. Please take a look at the general suggestions. And also mention (or even discuss) two further aspects: with the improving space based telescopes the exploration of exoplanets and their environmental modelling could also improve our knowledge. And mention as well the possiblity of laboratory and computer based simulations on the possiblities of exotic life forms. The referee encourages the authors to draw more conclusions than this simple summary. Try to provide discussion in the related suggested topics listed in the general suggestions above.

Author Response

REVIEWS AND RESPONSES TO Universe PAPER

Responses are in italics and bold

Reviewer 2:

First of all, let me congratulate to the authors to develop a manuscript in such a difficult topic, where the balance between theoretical - hypothetical argumentation and realistic, established scientific methods are diffiucult to find. As a results, I suppose they are expected to have strong critics from the referees - thus please consider my critical points as supportive and constructive comments.

The topic of the manuscript is important: the possiblity to identify some methodologial aspects what are like and how to search for the possilibity of life that is highly different from those we know on the Earth. Although it is quite difficult to approach such a theretical topic with a clear and well structured format, the manuscript could and should be improved further with these aims. The language of the manuscript is ideal, the structure is moderately good, however more information including figures or talbes that explain the possibilites and support their comparison would be useful. The referee suggests major revision and encourages the authors to work on the manuscript as it has potential and the topic is poorly discussed n the literature. Some general suggestions are listed in the general aspects below, however please consider carefully the specific suggestions too, as there are few (not many) quite important one also among them. An imprtant

General suggestions to improve the manuscript:

Although the citations cover a wide and representative part of the corresponding literature, some more and specific citations are necessary to improve the context.
And it is important in such a topic to indicate already at the beginning that astrobiology is doing a wide range of research in this domain even these are targeting the possiblit and detectability of Earth-like life (and cite: https://ui.adsabs.harvard.edu/abs/2016AsBio..16..561D/abstract)

This is noted, and citation is inserted at end of first paragraph of Introduction

At the definition of life (line 59) please cite Ganti's pioneering work, what was prouced early (Ganti, T. (1971). The Principle of Life (In Hungarian language). Budapest: Gondolat.) but became widely known only later: Ganti, T. (2003). The Principles of Life. Oxford University Press or Ganti, T. (2004). Chemoton Theory Vol I and II. Kluwer.

Gánti citation has been added in addition to some other important references

A possible improtant issue is the "ability for evolution" and its relation to reproduction. This is a theoretical question: if there is a potential life form that fulfil the main requrements listed in line 93 and later, but if the potential organism (or potential form of life) is able to reproduce but not able for mutation (or similar process), it will probably not evolve toward "higher level", and its classification as life is questionable. This aspect is partly touched in the lines 94-100 - however this might requre more discussion there.

Our paragraph points out that many forms of life are static (not evolving in a Darwinian sense) over extensive periods of time, but this doesn’t preclude their “being alive.”  We believe the current wording is sufficient.

It would be better to have a bit more discussion on LUCA in the point of view how its estimated characteritics might widen/sharpen our definition of life and possibltiy to identify alien life forms. It might improved any model apporach of alien life.

A clarifying sentence addressing this point has been inserted as the second sentence in the first paragraph of Section 5.

The section on Fermi paradox: this is a good intellectual adventure, however it would be much better to put less emphasis on Fermi paradox (and shorten this part) as there are more topics to be developed, which are closer related to this work (see the other general comments)

Discussion of the Fermi Paradox has been shortened (by deleting two larger paragraphs)

1080-1088 lines: this part is too much Solar System oriented. Please consider to expand to a wider outlook (as this part is the Discussion and Conclusion section) to extrapolate to exoplanets and exomoons

As the first sentence of that paragraph (lines 1074-5) indicates, planets and moons in our Solar System are cited as examples of different varieties of alien worlds than can be expected to be found throughout the Universe, so our argument using them as examples is generic – not focused on them alone.  The paragraph immediately following cites examples of planetary forms not found in our Solar System.  Therefore, we believe that further expansion is not necessary. However, we do discuss exoplanets explicitly in the last extended paragraph of new Section 8 (see also below).

It would be useful and almost necessary to provide some further conclusive aspects at the end of the work. In the current form it is an interesting collection of exotic possiblites, but does not provide specific findings. Althoug it is quite difficult to provide some specific conclusions of an almost theoretical work, but please consider, could the authors give further and useful conclusive remarks in the following aspect:
What type of further research topics could be focused on, based on the listed findings and suggestions (even if the conclusion is not a realistic plan for how simulation of alien life properties could be made, but this should be also noted) It would be interesting to see some theoretical "flow down" of arguments, what the discussed topics in the work suggest, regarding how to approach the problem, and is there observational possiblity? How widen the observed characteristics or their interpretation to make the observations based research more relevant for not "Earth-like life"?

A new Section 8 has been written to summarize the most plausible forms of life likely to be found on the different types of planetary habitats highlighted in this review. This new section includes conclusions about the observational possibilities for such worlds, and the lab experiments and simulations, and studies in analog environments on Earth, currently underway or planned.

Would it be possible to make a table with the topic groups listed in 3.1, 3.2, 3.3, 5.1, 5.2., 5.3 as theoretical possiblities in the lines, and columns on how to approach the given possiblity, could it be tested in laboratory or with computers, does the given life form has relevant consequences on its environment with changing it, is there some theoretical observatonal possiblitiy etc.?
For example based on the text between 481-497 lines, would it be possible to say something specific for observability (not about specific instrumentation but about what type of approach might be useful for the detection of such life forms)?
It might be also a possiblity to compare the listed hypothetic types of life according to their estimated ability to change their environment.

A table has been added in the new Section 8, as requested.

Specific aspects

39 line "then life on other worlds is likely to be very rare"
Do you mean life like on Earth? (based on water and organics)

We mean any form of life – if, as the preceding sentence states and the current sentence concludes, life as we know it is the only life possible.

62-66 lines: it is good to list many requirements, even if different groups are used in different approaches. One what I miss is to specifically mention of "such coding mechanism that provides heritage" - as the "reproduction" might not necessarily cover this.

(14) coding for retention of form and function across generations” has been added

64 line: please discuss briefly (even with some words in bracket), why homochirality is part of the definition of life and not a requirement / prerequisite for the origin of life (for chemical based life). These two aspects are close to each other, but not the same, thus would be good to mention, especially if you compare homochirality to the 2, 4, 9 items in your list.

This is a list of attributes invoked by various authors in their definitions of life, not of conditions necessary for life. Therefore, defense of any of these terms is not our intent, and we believe would clutter the discussion unnecessarily at this point.

66 line "teleology" is listed as one of the definitions, however there is no any reason mentioned for that, and the referee thinks it should not fit to this list of definitions. Please provide reason for that or delete. Even if this comes from a cited work, please discuss briefly why is it listed - many readars will lack (can not see) the reason to include this.

We agree that it doesn’t belong as necessary (or even relevant) to the definition of life, but it has been invoked by some notable scientists (e.g. Jacques Monod). Bear in mind that this is a list of frequently occurring attributes argued by others, not necessarily by us.

80-82 lines: please define what do you mean on "alien life" and on "Earth-like life"
“Alien life” is any form of life not found on Earth.  The wording has been changed to “life on other worlds”
96-97 lines: "First, it can only apply to multiple generations." I'm not sure in this. Processes like innovation in biological systems might happen in other ways than generations, in theory a system could improve itself without new generations. I suggest to consider and discuss this with more details. If you take a system that is considered to be "life", it should have reached this state from an abiotic state in some way. If you know other way than the evolution (in a wide sense) than please discuss this. But the contradiction (no evolution is necessary for all types of hypothetic life, but in this case how could emerge) should be solved. And the second part of the argumentation ("it implies change by natural selection, which even in some terran forms occurs barely at all over extended periods, and in alternative (e.g. mechanical) forms of life might not occur at all.") please formulate the sentence better than the current version.The paragraph has been rewritten to acknowledge what we take to be the reviewer’s concern.

112-113 lines at the end of the Introduction section: it would be useful to put one sentence on how do you discuss the problem area in the rest of the work and what is the logical structure behind.

The requested sentence has been added at the end of subsection 2.3.

120 line: "recurrent heavy" please define better, what do you mean? Early bombardment around the end of primary accretion plus LHB together? Or stochastic impacts or what?

 The term has been dropped and the section reworded for clarification
187 line: is there any summary, consequence or other final outcome that comes from this section? It ends a bit "abruply".

The wording of this final paragraph has been changed, to provide a more generic-sounding and less abrupt end to the section.

193-197 lines: interesting part, is there any theroetical consequence on the observabilit of such life forms?
There are no theoretical consequences we can think of, other than the idea that our usual notion of a living organism as an individual being of a consistent size and shape is not the only form of life, even on Earth, so we should expect the possibility of finding amorphous living entities on other worlds.

201-203 lines: interesting! would be good to have a bit more info here on this
There are not many examples of this life form, and they are not very well known.  We have cited the most relevant references that we could fine.

208-208 lines: also more info would be useful
Clonal plants are well known, and the aspen clones found across much of the northern hemisphere are good examples.  Our point here is simply that plants that appear to be individual organisms are in fact extensions of a single, continuous entity.

232 line: at ".planetary bodies, as already claimed for Mars [39, 42, 43]." please also cite this, relevant for next mission: https://ui.adsabs.harvard.edu/abs/2016OLEB...46..435K/abstract
The reference has been added, but in the new Section 8 where it more properly belongs in the discussion of robotic studies on Mars.

240-242 lines: a bit more information would be useful here.
A sentence providing more specific information has been added.

256-256 lines: "While structures like coral reefs would be easier to detect than other forms of life on other worlds because of their size" OK, but indicate if the observer has very high spatal resolution. Current attempts are more targeting specific spectral aspects than simply large scale features on a planetary surface. Your finding is relevant for the far future optical interferometers.

A parenthetic notation that an observer would have to have very high spatial resolution has been added.

270-272 lines: suggest to consider the so-called infra-biolgical systems (Fernando and Szathmáry 2009) also

A similar but more relevant paper co-authored by Szathmáry has been added

277 line: "Most of life on Earth has remained in its ancestral form, evolving little since its origin." Absolutely not clear, are you sure did you intend to wirte this?

The sentence has been amended to read, “Microbial life on Earth has remained essentially in its ancestral form . . . “

281 line: "requires preconditions that take time to develop" please give example
284-285 "create a complex new function" please give example, and consider here to cite: Szathmary, E. and Maynard-Smith, J. (1997). From replicators to reproducers: the first major transitions leading to life. Journal of Theoretical Biology, 187:555-571.

Examples of the different forms of transition have been inserted, and the reference to Szathmary and Maynard-Smith has been included at the most relevant point in the previous paragraph.

293 line "relatively soon," please indicate at least the scale of duration
Because evolutionary transitions differ in scope and antecedent conditions, no single definition of “relatively soon” is possible.  Industrial melanism in moths appeared in Europe within decades, while the evolution of wings from forearms undoubtedly required millions of years.

290-298 lines: please indicate is this text for surface or subsurface (or both) biospheres? And for chemo- or for photosynthesis?
The principles discussed in this paragraph apply to all biospheres and all metabolic pathways.

329 line: endosymbiosis should be a bit more detailed, especially regarding tis potential under "different" life forms or its role on the evolution of Earth-like life. Is there some furher theoretical possiblity, what could have been exploited by endosymbiosos but did not happen on Earth? This is a very difficult question I see, please skip if you have no relevant idea for the revised manuscript.
Since endosymbiosis is a Many Paths process, it clearly could apply to life elsewhere in the Universe, but absent knowledge of the cellular architecture of alien life forms, speculation about endosymbiosis in such forms of life is pointless. Since this section refers only to life on Earth, any discussion of life other than on Earth would be out of place anyway.

353 line: "can occur in very little time" please quantify, at least the scale of duration
The wording has been changed to “within a couple of days”.

358-362 line: please mention briefly the relevance of this on biomorphological structures
The parenthetical statement that “complex morphological innovations require more time. . . “ has been inserted

267 line: "oxyen" - it would fit well to the topic of your manuscript to indicate, what other "easy" andpotentially accessible energy source could be exploited by chemical oriented life forms
We assume the reviewer means line 367. The essence of bioenergetics for life on Earth is the oxidation of energy-rich molecules.  There is no real alternative to oxygen as the oxidizing agent (with the possible exception of fluorine, which is too scarce and too reactive).

361-372 lines: these arguments show that the effects emerged not as "external forces" but as an "internal mechanisms" by the interaction of different organisms - this might be mentioned, and also discussed somewhere that the components of the biophere migh exaggerate, spin up its own impovement and evolution

We are unclear here what the reviewer is referring to or wishes we should address

384-385 lines: "radically different environments from those in which terran life evolved" please present / formulate this differently, to make it more understandable
The sentence has been reformulated to state specifically the environment in which terran life evolved.

417 line: "Life as we don´t know it may" is the "it" necessary here?
Yes, this is preferred English syntax. The pronoun “it” refers to “life”, to make clear what is not known.

around the end of 5.1 section, would it be interesting to consider the followign possiblity? https://www.newscientist.com/article/dn12466-could-alien-life-exist-in-the-form-of-dna-shaped-dust/
A reference to this idea  has been inserted into the final paragraph of subsection 5.1.
356-360 lines (about there but all along the 5.4 section): do you consider possibility for natural emergence of such hypothetic machines? Or only by already evolved complex life (like those on Erth as technical civilisation)? Please indicte, this is not evint from the curent text.
We assume that mechanical forms of life would initially be fabricated by intelligent forms of organic life but eventually carried out robotically under the control of an advanced form of artificial intelligence. A statement to this effect has been inserted into the next-to-last paragraph of subsection 5.4..

575-584 lines: please cite as a discussion of various factors on habitability this work: https://ui.adsabs.harvard.edu/abs/2016OLEB...46..473K/abstract
The citation has been inserted at the beginning of subsection 6.1.

Figure 2: why geothermal heat is not indicated?
This figure only aims to model ionic and osmotic gradients as energy sources. The prior paragraph makes clear that chemical and geothermal sources of energy could also provide the energetic basis for an ecosystem.

633-634 lines: "the low level of available energy in ice-covered oceans" have you considered geothermal heat? In ideal case you should cite the corresponding reference or present some numerical comparison/argumentation why such subice oceans are "energy poor"? And also in what aspect? Relatively to the total habitable volume, compared to what?
Yes, we have considered in detail geothermal heat and a number of other energy sources, and explained why they would most likely result in subsurface oceans that are energy poor, in our previous publications, which are cited in the paragraph just prior to Fig. 2.

646-647 lines: "However, as Mars became colder and drier, its water became locked up either at its polar ice caps or in the subsurface" please cite: https://ui.adsabs.harvard.edu/abs/2012AsBio..12..586K/abstract

The citation has been added.

649-652 lines: "There is also evidence of temporary surface water, which, however, sublimates quickly when it erupts and leaves only a trail of salt [95]." please consider deliquescence already here: https://ui.adsabs.harvard.edu/abs/2015NatGe...8..357M/abstract and https://ui.adsabs.harvard.edu/abs/2020Icar..34013639P/abstractThe first reference already appears at the end of the paragraph.  The second reference has been added.

682 line: "their underside by colonies dominated by cyanobacteria" please cite https://ui.adsabs.harvard.edu/abs/2002IJAsB...1..305C/abstract
The citation has been added

703-706 lines: "a barren planetary body with no history of having had surface liquid is unlikely to have harbored the evolution of any form of life" do you mean subsurface habitability does not matter much here?
We do assume that life is most like to originate on the surface or near-surface, because interfaces provide much more likely platforms for constrained, complex chemistry.  But to avoid the implication that life could not evolve below the surface, the word “surface” has been removed from the sentence.

771 line: "Switzer (, 2011 #2165} showed" please provide proper citation
Citation has been corrected

817-825 lines: this part contains some poor text nd content, what should be develped further toward higher level, more sophisticated and more scientific parts. Arguing "For example, snake-type organisms" is not relevant, however a bit more established content might improve this part substantially. I would suggest focus on more general aspects and what is relevant for simpler organism (as the approach of the manuscript is to consider different possbile life forms, it should be broader than such very Earth-relevant specific items than "snake type"). Please consider the effect of stronger gravity on the verical characteistics of the atmosphere, on global nutrient migration, difference in redox gradient build up compared to lower gravity, restrictions in material circulation, subsurface void closure depth etc.

The paragraph has been rewritten in accordance with these suggestions.

830 line: "the Super-Earth has a large moon at a moderate distance," please explain why, suggest to consider that larger mass means larger inertia and larger stability of rotational parameters without a moon
Language to this effect has been added.

839-840 lines: OK, but consider these are relevant only for surface life
This qualification has been added.

861 line: "cannot host a biosphere" consider to modify to "unfavourable to host." and cite the corresponding work
The wording has been modified as suggested

863 line: "and if life exists at all it" do you mean "surface life"?

No, we mean surface or sub-surface life, since the moderate temperature at the terminator would extend beneath the surface to some depth.
around this above mentioned part it would be useful to discuss, could incrased irradiation (wether UV or ionizing radiation) support the build up of redox gradient, and suggest to cite and use this work: https://ui.adsabs.harvard.edu/abs/2018PrPNP.102....1L/abstract
This is interesting work, but in our opinion is too peripheral to the main point of this section.

884-889 lines (and around this part): it would be useful to discuss more the role of catastrophes, which can have favourable and unfavourable consequences for a biosphere
True, but again, beside the point of this section.

887-888 lines: "Evolution to more complex life is spurred by a highly heterogeneous environment [3] and seasonal variations largely contribute to that heterogeneity" OK, but please consider the abundance of available nutrients and energy sources, what do metter also
The abundance of nutrients and energy sources is implicit in the term “heterogeneous environment”.

889-890 lines: "most of them will also not be" consider to modify to "most of them might not be" and discuss briefly why
This clause has been removed.

902 line: "it may seem strange to consider the planetary atmosphere as a habitat" it would be good to provide some furhter arguments why it can be considered as a poor habitat, or at least why less favurable compared to others
The negative features of having a low buoyant density, sparse nutrients, and scarce liquid are cited as reasons for atmospheres as poor habitats.

919 line: "Venusian surface until about 700" consider to modify to "roughly about" as this 700 Ma is really a very rough estimation
Done

922 line "given the similar environmental" consider to modify to "possibly resmble" instead of similar
 We modified the wording to “likely” to acknowledge the reviewer´s comment und general uncertainty even though the information we do have from the early environmental conditions of both Venus and Earth show many  similarities

934-947 lines and around here in this work: please try fo put more emphasis on exoplanets (and less is enough from Venus then currently discussed)
Almost nothing is known about the atmospheres of any exoplanets. Venus is an accessible prototype and thus worthy of consideration for both theoretical and practical reasons.

963 line: "which do not occur on gas giants" maybe, but it could be mentioned that this environment might be also rich in solid-liquid-gas interations in microscopic scales if there are many condensation nuclei and strong vertical mixing
This possibility is added parenthetically.

1035-1039 lines: this is a hypothetical argumentation, to state this, numercal calculations would be necessary - what goes beyond of the scope of this work. I suppose it should be mentioned that this paragraph is based on "simple theoretical argumentation"
We believe that the hypothetical nature of the argument is obvious with the current wording.

1061 line "self-replicating machines" and earlier related parts: please emphasize that the origin of such machines requires other forms of life to produce them (even if they might be self standing later), what could be taken as a unique aspects among the other forms of life, which are all have natural origin. It would be a too far reaching discussion to discuss the difference between naturally originated life and "artificially produced" life- however would be useful to note this.
This point was made earlier, in the next-to-last paragraph of Section 5.

1082 line: "even more advanced forms in sequestered habitats like lava tube caves" please note that lava caves on Mars might be relatively small (compared to other habitats) and need not be connectedor fewmight be connected, what might not be "enough" for the emergence of advanced living organims
Lava tube caves on Earth are large, and visibly larger on Mars due to the lower gravity.  In any event, the size of the habitat should not be a restriction for small animals and amorphous forms of life like microbial mats and fungi, which need not have arisen inside the caves.

1096 line: "gas giants present environments almost certainly too harsh" the referee still not feel that this statement became well established in the earlier parts of the manuscript. Could you consider the wide range of gas giants might be present among exoplanets? As they cool while getting oder, range of condensation can happen inside them, what together with various mixing processes might make interesting environments for the origin of life - or might not make. There also migth be liquid surfaces in the interior. But to state they are harsh is not established or proven in this manuscript.
The data we have on the atmospheres of Jupiter (from the Galileo probe) and Saturn (from the Cassini probe) clearly suggest that temperatures and pressures with increasing depth (but still in the gaseous phase) are too great for the survival of any forms of life found on Earth.  However, as the paragraph later states, the upper clouds of the gas giants conceivably could support life.  While condensations in older gas giants conceivably could create liquid-gas interfaces that might increase the possibility that like could be sustained in them, the problems of how the life could originate and how it could harvest sufficient nutrients or other forms of energy from the clouds would remain.  The possibility for life in the clouds of the gas giants admittedly cannot be ruled out, but to do such a theoretical possibility justice would require elaboration well beyond the scope of this review.

1105 line "it is premature." yes, but this is required to plan instruments and run research project. Please indicate this also.
The purpose of this sentence is to point out that a generic “habitable zone” based on assumptions of terran life only, is too restrictive for imagining non-terran forms of life.  It is not intended as a criterion for planning instruments or running research projects.

After 1112 line: I miss here more discussion. Please take a look at the general suggestions. And also mention (or even discuss) two further aspects: with the improving space based telescopes the exploration of exoplanets and their environmental modelling could also improve our knowledge. And mention as well the possiblity of laboratory and computer based simulations on the possiblities of exotic life forms. The referee encourages the authors to draw more conclusions than this simple summary. Try to provide discussion in the related suggested topics listed in the general suggestions above.

We thank the reviewer for these suggestions, and believe they are now well covered in the new Section 8, which also includes the added detailed Table 1 addressing the reviewer´s concerns above.

Reviewer 3 Report

Interesting review. It would be appropriate to add the following references:

Luisi PL (1998) About Various Definitions of Life. Orig Life Evol Biosph 28: 613-622.

Bains W (2004) Many Chemistries Could Be Used to Build Living Systems. Astrobiology 4:137-167

Ward PD, Benner SA (2007) Alien Biochemistries. In Planets and Life, (Sullivan III WT and Baross JA, Eds.) Cambridge University Press, pp 537-544.

Author Response

REVIEWS AND RESPONSES TO Universe PAPER

Responses are in italics and bold

Reviewer 3.

Interesting review. It would be appropriate to add the following references:

Luisi PL (1998) About Various Definitions of Life. Orig Life Evol Biosph 28: 613-622.

Reference added in definition of life section

Bains W (2004) Many Chemistries Could Be Used to Build Living Systems. Astrobiology 4:137-167

Reference included in Section 5

Ward PD, Benner SA (2007) Alien Biochemistries. In Planets and Life, (Sullivan III WT and Baross JA, Eds.) Cambridge University Press, pp 537-544.

Reference included in Section 5

Round 2

Reviewer 1 Report

The paper is, now, almost ready for acceptance. Nonetheless, the authors must cite the very recent

Keiichi Wada, Yusuke Tsukamoto, Eiichiro Kokubo,

Formation of "Blanets" from Dust Grains around the Supermassive Black Holes in Galaxies

arXiv:2007.15198 [astro-ph.EP]

2020

and

L. Iorio, Effects of the general relativistic spin precessions on the habitability of rogue planets orbiting supermassive black holes, Astrophys. J. 896 (2020) 82

which was already mentioned in previous report.

Author Response

REVIEWER FEEDBACK-V2

Reviewer 1:

The paper is, now, almost ready for acceptance. Nonetheless, the authors must cite the very recent

Keiichi Wada, Yusuke Tsukamoto, Eiichiro Kokubo, (Formation of "Blanets" from Dust Grains around the Supermassive Black Holes in Galaxies arXiv:2007.15198 [astro-ph.EP] 2020), and
L. Iorio, Effects of the general relativistic spin precessions on the habitability of rogue planets orbiting supermassive black holes, Astrophys. J. 896 (2020) 82, which was already mentioned in previous report.

The earlier reference to Wada et al. has already been cited (#124) and is more appropriate.  The newer reference to Iorio suggested here has been added (#126).

Reviewer 2 Report

General aspects

The authors improved the manuscript, considered many of the referee’s comments. The manuscript is close to acceptance, and beside few minor aspests (listed later), there is only one but major issue: The authors still keep listing among the cited definitions of life in line 68 „teleology”. Althought in the authors indicated there is a publication on it as a defintion, there is not scientifically based reasons or established argumentation behind. The authors indicate they might agree in this aspect with the referee, however they did not delete from the revised version explaining: „We agree that it doesn’t belong as necessary (or even relevant) to the definition of life, but it has been invoked by some notable scientists (e.g. Jacques Monod). Bear in mind that this is a list of frequently occurring attributes argued by others, not necessarily by us”.

A nice aspect of science is it does not matter who said a given statement, only that does matter is it true or could be proven by evidences. I suppose it can not be a reason that „said by a notable scientist”. Teleology in that work was not proven or demonstrated to be necessary or even established by scientific reasons. The authours of this current manuscript should keep in mind that they have responsibility who to cite, and later each work (like this one they are working on) influences the general view of the scientific community. The measure have to be always the scientific excellence and the testability, the rational reasoning with proven by evidences. As this is not the case with „teleology”, and if the authors of this current work consider their manuscript as established on scientific disciplines, they should not include „teleology” but delete it. It can not be a reason that it has been published somewhere – many fake works have been also published, thus the authors should select carefully who they cite as they are (as everybody) responsible for their work.

Specific apects

subtitle changes improved the mauscript

at the referee’s earlier comment:

„267 line: "oxyen" - it would fit well to the topic of your manuscript to indicate, what other "easy" andpotentially accessible energy source could be exploited by chemical oriented life forms
We assume the reviewer means line 367. The essence of bioenergetics for life on Earth is the oxidation of energy-rich molecules.  There is no real alternative to oxygen as the oxidizing agent (with the possible exception of fluorine, which is too scarce and too reactive).

Yes, and suggest to mentond this: put into the manuscript somewhere that fluorine might have been similar but it is too scarce and too reactive.

341 line: „low-fitness periods[65].”

put a space before the citation

430 line „(-94oC to 65oC under”

correct the „oC” symbol

745-757 lines: good new paragraph, but you might also mention planets around pulsars or neutron stars. And also consider in a heavy irradiated environment 100-200 m below the surface of a rocky planet the conditions need not necessarily be inhospitable or should be analysed and evaluated – actually this has not been deeply considere by the knowledge of the referee.

Thank you for the table, I think the readers will like it! At the 4th line from the bottom, second column, after „Ceres” I would put a question mark in braket as it is less certain does or did Ceres has water, compared to Europa of Enceladus. Consider: „Ceres (?)”. And the „Triton” case (before the last line) is not clear for the referee, what did you intend to indicate here, what conditions. On rogue do you mean very cold? On Triton there is an actve surface and subsurface environment with cryovolcanism, and moderately young surfaces, „geysers” etc.

1032 line „in all habitats” put a dot at the end of the sentence.

1047-1048 lines: „returning valuable data on the physicochemical characteristics of their clouds”, yes but consider to expand with: „…characteristics of their clouds – however more data is necessary to estimate their astrobiology potential.”

1049 line „Ceres, Europa, and Enceladus, among others” consider to modify to „Europa, Enceladus and possibly Ceres, among others”

Author Response

Reviewer 2 (our responses in italic):

General aspects

The authors improved the manuscript, considered many of the referee’s comments. The manuscript is close to acceptance, and beside few minor aspests (listed later), there is only one but major issue: The authors still keep listing among the cited definitions of life in line 68 „teleology”. Althought in the authors indicated there is a publication on it as a defintion, there is not scientifically based reasons or established argumentation behind. The authors indicate they might agree in this aspect with the referee, however they did not delete from the revised version explaining: „We agree that it doesn’t belong as necessary (or even relevant) to the definition of life, but it has been invoked by some notable scientists (e.g. Jacques Monod). Bear in mind that this is a list of frequently occurring attributes argued by others, not necessarily by us”.

A nice aspect of science is it does not matter who said a given statement, only that does matter is it true or could be proven by evidences. I suppose it can not be a reason that „said by a notable scientist”. Teleology in that work was not proven or demonstrated to be necessary or even established by scientific reasons. The authours of this current manuscript should keep in mind that they have responsibility who to cite, and later each work (like this one they are working on) influences the general view of the scientific community. The measure have to be always the scientific excellence and the testability, the rational reasoning with proven by evidences. As this is not the case with „teleology”, and if the authors of this current work consider their manuscript as established on scientific disciplines, they should not include „teleology” but delete it. It can not be a reason that it has been published somewhere – many fake works have been also published, thus the authors should select carefully who they cite as they are (as everybody) responsible for their work.

With respect, we cannot remove “teleology” from the list of characteristics that have been associated with life, because as a factual matter, teleology has been associated with “life” for centuries.  It was taken for granted by Plato, Aristotle, Kant, Marx, and Bergson. In the 19th century, evolutionary theorists like Jean Lamarck and Alfred Wallace embraced it implicitly, and in the 20th century, the molecular biologist, Jacques Monod, and the paleontologist, Teilhard de Chardin, invoked it.  We agree with the reviewer that it has no scientific support, though it is a theoretical possibility.  One of us (LI) has co-authored a book on “The Evolutionary Imperative”, which considers the question of whether there is a natural explanation for the apparent directionality of evolution (toward increased complexity and diversification).  This is all that “teleology” means – that there is an inherent (not necessarily supernatural) drive toward a particular endpoint. Again, we agree that it does not belong as a necessary attribute in the definition of life, and we state in the next-to-last sentence of that paragraph that its inclusion is controversial. But it has been considered an important element of a theory of life by serious thinkers.  That is all that the paragraph is pointing out.

Another reason the term belongs on this list in particular in a review aimed at an audience of astrobiologists is that a  number of astrophysicists, astronomers, and cosmologists (unlike biologists) within the astrobiological community often take seriously the anthropic cosmological principle, which (especially in the strong version) is clearly a teleological concept.

Specific apects

 subtitle changes improved the mauscript
at the referee’s earlier comment:

„267 line: "oxyen" - it would fit well to the topic of your manuscript to indicate, what other "easy" andpotentially accessible energy source could be exploited by chemical oriented life forms
We assume the reviewer means line 367. The essence of bioenergetics for life on Earth is the oxidation of energy-rich molecules.  There is no real alternative to oxygen as the oxidizing agent (with the possible exception of fluorine, which is too scarce and too reactive).” Yes, and suggest to mentond this: put into the manuscript somewhere that fluorine might have been similar but it is too scarce and too reactive.

      The suggested wording has been inserted in lines 358-9..

341 line: „low-fitness periods[65].”  put a space before the citation

      A space has been inserted before the citation

430 line „(-94oC to 65oC under”  correct the „oC” symbol

      “oC” has been corrected to “ oC” in line 430

745-757 lines: good new paragraph, but you might also mention planets around pulsars or neutron stars. And also consider in a heavy irradiated environment 100-200 m below the surface of a rocky planet the conditions need not necessarily be inhospitable or should be analysed and evaluated – actually this has not been deeply considere by the knowledge of the referee.

Mention of pulsars has been inserted, along with the possibility of habitability at great depth on a rocky planet in a high radiation environment, in paragraph between lines 745 and 756.

Thank you for the table, I think the readers will like it! At the 4th line from the bottom, second column, after „Ceres” I would put a question mark in braket as it is less certain does or did Ceres has water, compared to Europa of Enceladus. Consider: „Ceres (?)”. And the „Triton” case (before the last line) is not clear for the referee, what did you intend to indicate here, what conditions. On rogue do you mean very cold? On Triton there is an actve surface and subsurface environment with cryovolcanism, and moderately young surfaces, „geysers” etc.

A question mark has been added after Ceres. Triton is listed in the table as a Solar System body which most likely is a captured rogue planet.  It does happen to be extremely cold, as we assume most rogue planets would be, though the nature of their origin, whether they have an atmosphere, their mass and geophysical activity, and other factors would determine their habitability (what we mean by “Dependent on surface, atmosphere, and planetary history”).

 1032 line „in all habitats” put a dot at the end of the sentence.

        A period has been placed at the end of the sentence.

1047-1048 lines: „returning valuable data on the physicochemical characteristics of their clouds”, yes but consider to expand with: „…characteristics of their clouds – however more data is necessary to estimate their astrobiology potential.”

That more data is necessary to estimate their astrobiological potential is an obvious statement, and therefore an unnecessary addition – especially since the paragraph deals only with technological capabilities, not evaluation of habitability.

1049 line „Ceres, Europa, and Enceladus, among others” consider to modify to „Europa, Enceladus and possibly Ceres, among others”

Such a change in wording would suggest that Ceres is only “possibly” reachable by current technology.  Since a probe from Earth has already landed on Titan, Ceres is certainly, not “possibly,” reachable by current technology.